# Simulations approaching data: cortical slow waves in inferred models of the whole hemisphere of mouse

Cristiano Capone [1,8✉], Chiara De Luca[1,2,8], Giulia De Bonis [1], Robin Gutzen[3,4], Irene Bernava[1], Elena Pastorelli [1], Francesco Simula[1], Cosimo Lupo [1], Leonardo Tonielli[1], Francesco Resta[5], Anna Letizia Allegra Mascaro [5,6], Francesco Pavone[5,7], Michael Denker[3] & Pier Stanislao Paolucci [1]

The development of novel techniques to record wide-field brain activity enables estimation of data-driven models from thousands of recording channels and hence across large regions of cortex. These in turn improve our understanding of the modulation of brain states and the richness of traveling waves dynamics. Here, we infer data-driven models from high-resolution in-vivo recordings of mouse brain obtained from wide-field calcium imaging. We then assimilate experimental and simulated data through the characterization of the spatio-temporal features of cortical waves in experimental recordings. Inference is built in two steps: an inner loop that optimizes a mean-field model by likelihood maximization, and an outer loop that optimizes a periodic neuro-modulation via direct comparison of observables that characterize cortical slow waves. The model reproduces most of the features of the non-stationary and non-linear dynamics present in the high-resolution in-vivo recordings of the mouse brain. The proposed approach offers new methods of characterizing and understanding cortical waves for experimental and computational neuroscientists.

[1] INFN, Sezione di Roma, Rome, Italy. [2] PhD Program in Behavioural Neuroscience, "Sapienza" University of Rome, Rome, Italy. [3] Institute of Neuroscience and Medicine (INM-6) and Institute for Advanced Simulation (IAS-6) and JARA-Institute Brain Structure-Function Relationships (INM-10), Jülich Research Centre, Jülich, Germany. [4] Theoretical Systems Neurobiology, RWTH Aachen University, Aachen, Germany. [5] European Laboratory for Non-Linear Spectroscopy, Sesto Fiorentino, Italy. [6] Neuroscience Institute, National Research Council, Pisa, Italy. [7] University of Florence, Physics and Astronomy Department, Sesto Fiorentino, Italy. [8] These authors contributed equally: Cristiano Capone, Chiara De Luca. ✉email: cristiano0capone@gmail.com

In the last decade, macroscale wide-field imaging, coupled with fluorescent activity indicators (e.g., Genetically Encoded Calcium Indicators, GECIs[1–4]), has provided new insights on the study of brain activity patterns[5–13]. Although this microscopy technique does not reach single-cell resolution and supports temporal sampling rates much lower than classical electrophysiology, it enables the recording of neural dynamics simultaneously across brain areas, with signal-to-noise ratio and spatiotemporal resolution high enough to capture the cortex-wide dynamics in anesthetized and behaving animals[14,15]. Furthermore, this technique allows to map spontaneous network activity and has recently provided important information about the spatio-temporal features of slow waves[11,12,16,18]. In this work we consider raw imaging datasets made up of 10.000 pixels per frame at 50 μm × 50 μm spatial resolution and sampled every 40 ms. This poses the challenge to build a model capable of being descriptive and predictive of this large amount of information.

The building of dynamic models requires to capture in a mathematical form the causal relationships between variables required to describe the system of interest; the value of the model is measured by both its ability to match actual observations and its predictive power. We mention two possible approaches to model building, the classical *direct* and the more recent *inverse*. In the former, parameters appearing in the model are assigned through a mixture of insights from knowledge about the main architectural characteristics of the system to be modelled, experimental observations, and trial-and-error iterations. In the latter, instead, model parameters are inferred by maximising some objective function, e.g., likelihood, entropy, or similarity defined through some specific observable (e.g., functional connectivity). A small number of models following the inverse approach have been so far able to reproduce, from a single recording, the complexity of the observed dynamics. In[19] the authors constrain the model dynamics to reproduce the experimental spectrum. The work presented in[20] proposes a network of modules called epileptors, and infers local excitability parameters. Similarly, the authors in[21] estimate parameters and states from a large dataset of epileptic seizures. Some models focus on fMRI resting state recordings and on the reproduction of functional connectivity[22]. However, few works center on the reliability of the temporal and spatial features generated by the inferred model, such as in[23], where spatio-temporal propagation of bursts of a culture of cortical neurons is accurately reproduced with a minimal spiking model. Indeed, the assessment of this kind of results on large datasets requires methods capable of extracting and characterising the emerging activity.

A major difficulty in assessing the goodness of inferred models is the impossibility of directly comparing the set of parameters of the model ("internal") with the set of those observable in the biological system, and the two sets are typically not coincident. In order to fill the gap between experimental probing and mathematical description, this paper proposes a method relying on a set of flexible observables capable to optimally perform model selection and to validate the quality of the produced prediction. The developed modular analysis procedure is able to extract relevant observables, such as the distributions of frequency of occurrence, speed and direction of propagation of the detected slow waves.

Another common problem when inferring the model parameters is that it is possible to only partially constrain the dynamics. Usually, the average activity and the correlations between different sites[24,25] are used. However, it remains difficult to constrain other aspects of the spontaneously generated dynamics[26], e.g., the variety of shapes of traveling wavefronts and the distribution of frequency of slow waves. Aware of this issue, we propose a two-step approach that we name *inner loop* and *outer loop*. In the inner loop, the model parameters are optimised by maximising the likelihood of reproducing the dynamics displayed by the experimental data locally in time (i.e., the dynamics at a given time is dependent only on the previous iteration step). Conversely, in the outer loop, the outcome of the simulation and the experimental data are compared via observables based on the statistics of the whole time course of the spontaneously generated dynamics. Specifically, we demonstrate that the inclusion of a time-dependent acetylcholinic neuro-modulation term in the model enables a better match between experimental recordings and simulations, inducing a variability in the expressed dynamics otherwise stereotyped. This additional term influences the excitability of the network along the simulation time course. The outer loop thus enables a quantitative search for optimal modulation.

Another aspect to be considered is the combination of a large number of parameters in the model and a typically short duration of recording sessions (in our case, only six recordings lasting 40 s each). Since the resulting inferred system could be underdetermined, it is important to use good anatomical and neurophysiological priors. Following this route, a minimal anatomical assumption, is to decompose structural connectivities into a short-range lateral connectivity contribution (the exponential decay characterising the lateral connectivity[27], or intra-area connectivity) and a long-range white matter mediated connectivity. It is well known, and confirmed in our experimental data, that during deep sleep and deep and intermediate anesthesia, propagation is by slow wave propagation and therefore can be mainly mediated by lateral cortical connectivity (ref. [28]). Relying on this, we chose to model lateral connectivity using local elliptic kernels. Elliptical kernels are often used in neural field theories to predict wave properties as a function of the connectivity parameters[29,30]; this is usually referred to as a direct approach, going from parameters to dynamics. In[31] the authors propose equations that include several properties of the cerebral tissue (2D structure, temporal delays, nonlinearities and others), retaining mathematical tractability. This allows to analytically predict global mode properties (such as stability and velocity of propagating wavefronts) for different geometries. Here, we propose a methodology to implement an inverse approach, from wave properties to model parameters. The choice of local connectivity kernels reduces the number of parameters to be inferred from $N^2$ to $N$ (number of recording sites). The proposed approach prevents overfitting and keeps the computational cost of inference under control even for higher-resolution data. In summary, this paper addresses the understanding of the mechanisms underlying the emergence of the spatio-temporal features of cortical waves leveraging the integration of two aspects: the knowledge coming from experimental data and the interpretation gained from simulations. We identified essential ingredients needed to reproduce the main modes expressed by the biological system, providing a mechanistic explanation grounded in the neuro-modulation and spatial hetereogeneity of connectivity and local parameters.

In the following, the Results section presents the main elements of this work (for methodological details, see Methods section) and the Discussion section illustrates limitations and potentials of the approach we introduced, in particular in relation with experiments, theoretical and modeling perspectives.

## Results

We propose a two-step procedure to reproduce, in a data-constrained simulation, the spontaneous activity recorded from the mouse cortex during anesthesia. Here, we considered signals collected from the right hemisphere of a mouse, anesthetized with a mix of Ketamine and Xylazine. The activity of excitatory

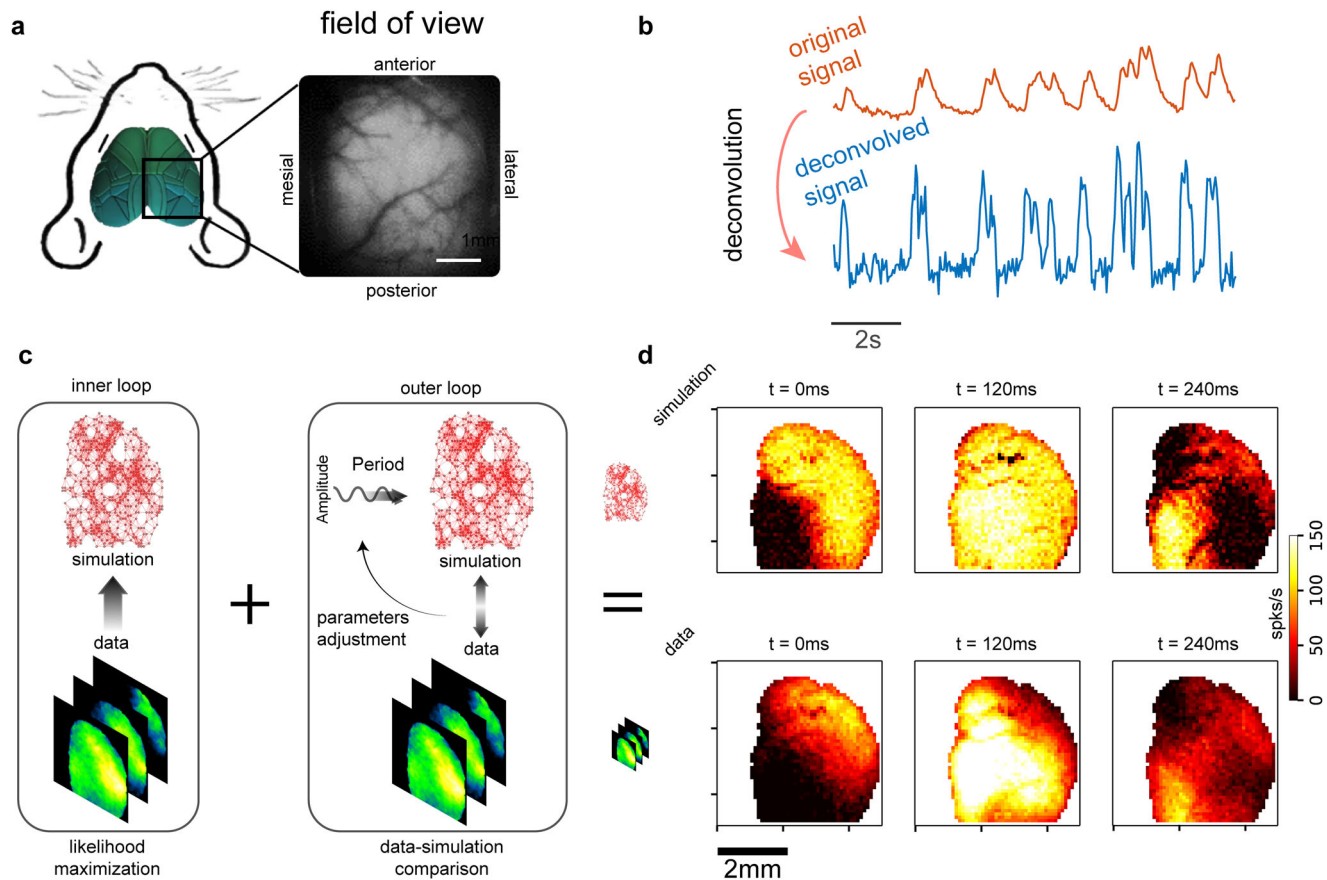

**Fig. 1 Experimental setup and sketch of the model. a** The cortical activity of the right hemisphere of a mouse (under Ketamine/Xylazine anesthesia) is recorded through wide-field calcium imaging (GCaMP6f indicator). **b** The calcium indicator response is slow if compared with the time scales of neuron activity. Applying to the original signal (upper) a proper deconvolution function fitted from single-spike response[7,79], it is possible to estimate a good proxy for the average activity of the local population of excitatory neurons recorded by a single pixel (bottom). **c** Sketch of the inference model, made up of an inner loop (likelihood maximization) and an outer loop (search for optimal neuro-modulation-related hyper-parameters granting the best match between simulations and data). **d** Three frames from simulation (top) and experimental data (bottom), both expressing slow wave propagation.

neurons labeled with the calcium indicator GCaMP6f is recorded through wide-field fluorescence imaging, see Fig. 1a. In this anesthetized state, the cortex displays slow waves from ≃0.25 Hz to ≃5 Hz (i.e., covering a frequency range that corresponds to the full δ frequency band and extends a little towards higher frequencies). The waves travel across distant portions of the cortical surface, exhibiting a variety of spatio-temporal patterns.

The optical signal is highly informative about the activity of excitatory neurons. However, the GCaMP6f indicator, despite being "fast" compared to other indicators, has a slow response in time if compared to the time scale of the spiking activity of the cells. The impact on the observed experimental signature can be described as a convolution of the original signal with a low-pass transfer function, Eq. (2), resulting in a recorded fluorescence signal that exhibits a peak after about 200 ms from the onset of the spike and a long-tailed decay[7] (see Methods, Experimental Data and Deconvolution section for details). To take into account this effect, we performed a deconvolution of the fluorescence signal (Fig. 1b–top) to better estimate the firing rate of the population of excitatory neurons, i.e., the multiunit activity, recorded by each single pixel (see Fig. 1b–bottom).

We considered two sets of six acquisitions (each one lasting 40 s) collected from two mice. We characterized the slow waves activity defining a set of macroscopic local observables describing the properties of the cortex site, in opposition to cellular properties derived from the inference: local speed, direction, and inter-wave interval (IWI, see Methods for more details). Also, we used these observables to compare the spontaneous dynamics measured in experimental recordings with the one reproduced in simulated data. In the first step of the proposed method, the inner loop, the parameters of the model are inferred through likelihood maximization, Fig. 1c–left, exploiting a set of known, still generic, anatomical connectivity priors[27] and assuming that mechanisms of suppression of long-range connectivity are in action during the expression of global slow waves[28]. Such anatomical priors correspond to probabilities of connection among neuronal populations, and assume the shape of elliptical kernels exponentially decaying with the distance. For each pixel, the inference procedure has to identify the best local values for the parameters of such connectivity kernels. In addition, for each acquisition period $tn \in \{t1, \ldots, t6\}$, it identifies the best local (i.e., channel-wise) values for spike-frequency-adaptation strength and external current.

In the second step, the outer loop, we seek for hyper-parameters (through a grid search exploration) granting the best match between model and experiment by comparing simulations and data, Fig. 1c–right. This second step exploits acquired knowledge about neuro-modulation mechanisms in action during cortical slow waves expression, implementing them as in[32].

In Fig. 1d, we provide a qualitative preview of the fidelity of our simulations by reporting three frames from the simulation (top) and the data (bottom). This example shows similarities in the

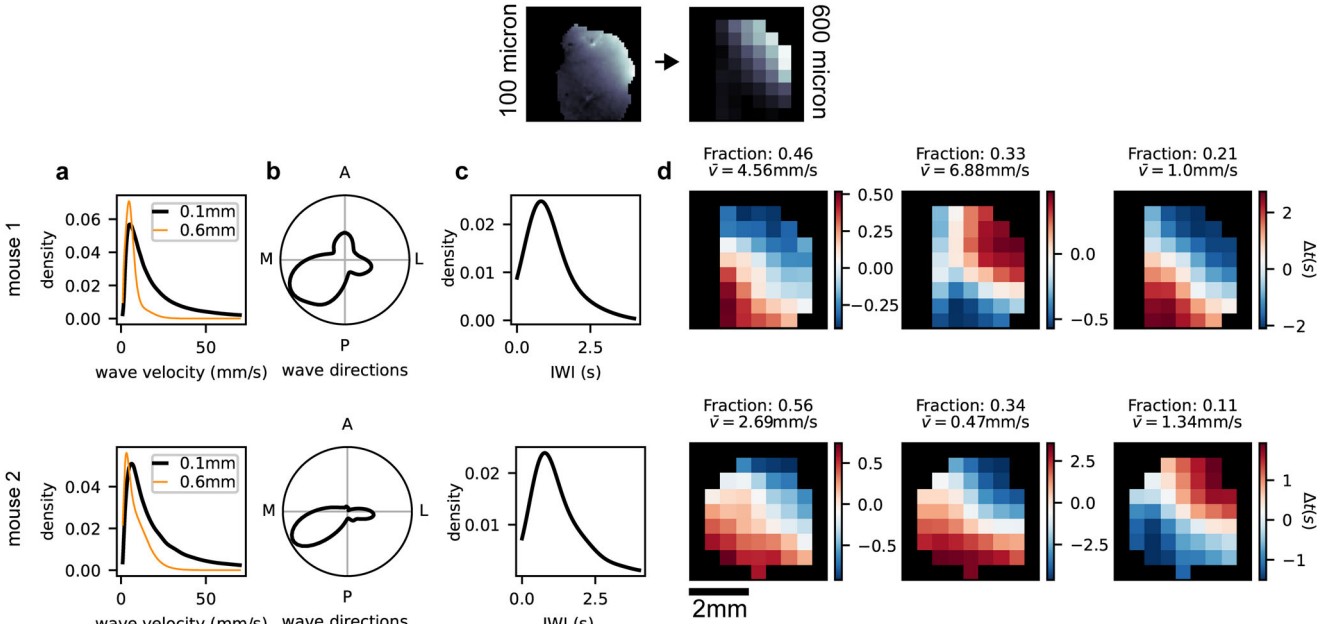

**Fig. 2 Summary of wave propagation properties in experimental trials for two different mice.** Measures for the quantitative characterization and comparison of waves accumulated across all pixels and all trials in mouse 1 (upper row) and 2 (lower row), respectively. **a** Local wave velocity distributions measured over the original dataset at a spatial resolution of 0.1 mm (black thick line) and the one downsampled at a spatial resolution of 0.6 mm (orange thin line). **b** Local wave direction distributions. **c** Local inter-wave interval distributions. **d**. Wave propagation modes identified by Gaussian Mixture Model (GMM). The GMM classification algorithm returns the optimal number of clusters appropriate for grouping the waves identified in the observed dataset, then assigning each wave to a cluster, assuming each cluster is described by a multivariate Gaussian distribution in the channel space (with its mean vector and covariance matrix). Distributions in panels **a**, **b** and **c** are at the original resolution (100 $\mu$m), while before application of GMM the channel-space dimension has been reduced to 600 $\mu$m (see the original resolution compared to the downsampled one on top of the main panels). For comparison, in panel **a** the velocity distribution is also reported for the downsampled signal. In the experimental data, three different wave propagation modes for each mouse are identified. The spatio-temporal propagation dynamics of each mode is normalized to the temporal center of each wave (from blue to red pixels).

slow waves propagation: the origin of the wave, the wave extension and the activity are qualitative comparable, as well as the activation pattern. A quantitative comparison is detailed in the following sections.

**Characterization of the cortical slow waves.** In this work, we improved the methodology implemented by[16] (see Methods for details), providing three local (i.e., channel-wise) quantitative measurements to be applied in order to characterize the identified slow waves activity: local wave velocity, local wave direction, and local wave frequency. The high spatial resolution of the data allows not only to extract global information for each wave, but also to evaluate the spatio-temporal evolution of the local observables within each wave.

For each channel participating in a wave event, the local velocity is evaluated as the inverse of the absolute value of its passage function gradient (see Eq. (21)), the local direction is computed from the weighted average of the components of the local wave velocities (see Eq. (22)), and the local wave frequency is expressed through its reciprocal, the inter-wave interval (IWI) evaluated from the duration between two consecutive waves passing through the same channel (see Eq. (23)).

In Fig. 2a–c we show a summary of the data analysis performed on all the recordings from the two mice. Interestingly, the defined experimental observables are remarkably comparable among the two subjects, since they display similar distributions in detected waves velocity, directions and IWI.

Further, we classified the waves into propagation modes with a Gaussian Mixture Model (GMM) classification approach applied in a downsampled channel space (44 and 41 informative channels for the two mice, respectively). The number of modes fitted by the

GMM is automatically identified using the Bayesian Information Criterion (BIC) relying on a likelihood maximization protocol[33]. For additional details see Supp. Mat. section Detecting the number of components in the GMM. In Fig. 2d we report the wave propagation modes identified through GMM (see details in Methods, Gaussian Mixture Model section). Specifically, GMM are fitted on a dataset composed by the observed spatio-temporal wave patterns represented in the subsampled channel space (i.e. for each entry in the dataset, the number of features is equal to the number of reduced channels). See section Methods, Slow Waves and Propagation analysis for a more detailed description of how travelling waves are identified from the raw signal. Coherently with what expected from previous studies[34], in both mice two main propagation directions are identified: a more frequent antero-posterior and a less frequent postero-anterior direction.

It is also worth noting that, as quantitatively shown in[35], the distributions of wave quantitative observables change as a function of the downsampling factor. With a decreasing spatial resolution, fewer waves are detected, and they appear more planar as some complex local patterns are no longer detected. In Fig. 2a, we show the distribution of local velocities measured over the original dataset at a spatial resolution of 0.1 mm (black) and the one downsampled at a spatial resolution of 0.6 mm (orange). Here, the distributions are sharpened and have a lower mean, coherent with the global velocities of the typical waves identified.

**Two-step inference method**

*The inner loop.* The first step of our inference method can be seen as a one-way flow of information from data to model and consists

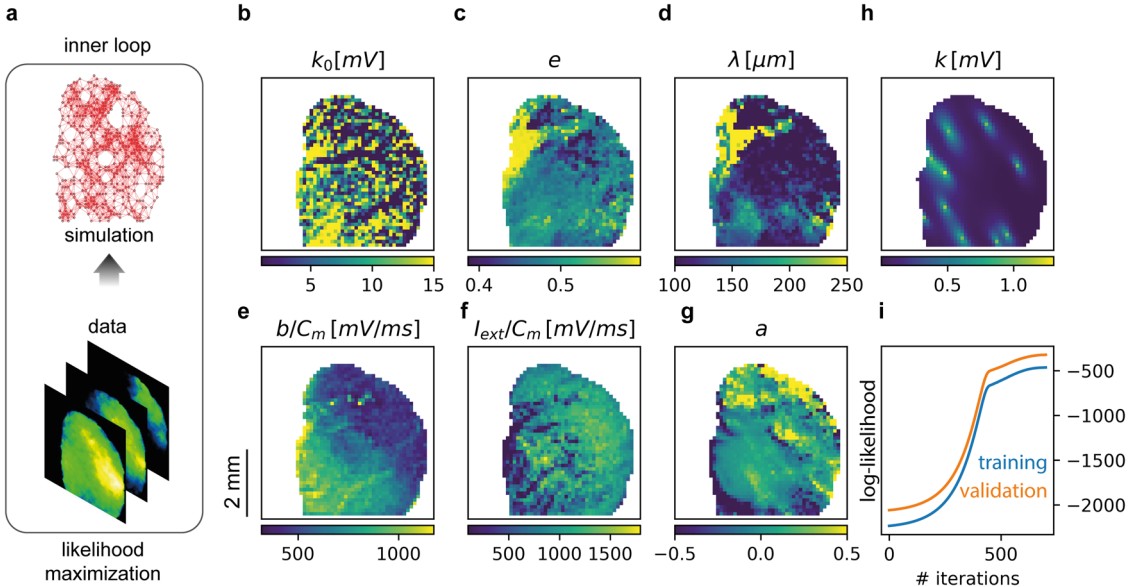

**Fig. 3 Inner Loop. a** The inner loop: likelihood maximization. **b–g** Spatial maps of estimated parameters after 700 iterations of iRprop[80], that we further improved by setting priors for the expected spatial decay law of lateral connectivity. Parameters $k_0$, $e$, $\lambda$, and $a$ are the connectivity parameters (weight, eccentricity, spatial scale, and anisotropy), as defined in Methods; $I_{ext}$ and $b$ are the local external input and the strength of the spike frequency adaptation. **h** Ten inferred elliptic connectivity kernels $k$, presented in an overlayed manner, that illustrate the existence of strong eccentricity $e$ for the elliptic connectivity kernels. **i**. Training and validation log-likelihoods (in blue and orange, respectively) as functions of the number of iterations.

**Table 1 Neuronal parameters defining the two-population regular and fast-spiking populations model.**

|  | $\theta$ (mV) | $\tau_m$ (ms) | $C_m$ (nF) | $V_r = E_l$ (mV) | $\Delta V$ (mV) | $\tau_i$ (ms) | $E_i$ (mV) | $b$ (nA) | $\tau_W$ (s) |
|---|---|---|---|---|---|---|---|---|---|
| exc | −50 | 20 | 0.2 | −65 | 2.0 | 5 | 0 | 0.005 | 0.5 |
| inh | −50 | 20 | 0.2 | −65 | 0.5 | 5 | −80 | 0 | 0.5 |

of estimating model parameters by a likelihood maximization, see Fig. 3a.

We built a generative model as a network of interacting populations, each of them composed of 500 AdEx (Adaptive Exponential integrate-and-fire) neurons. Each population is modeled by a first-order mean-field equation[29,36–38], according to which all the neurons in the population are described by their average firing rate. The neurons are affected by spike frequency adaptation, that is accounted for by an additional equation for each population (see Methods for details). We remark that we considered current-based neurons; however, the mean-field model might be extended to the conductance-based case[39,40]. Each population models the group of neurons recorded by a single pixel (the number of considered pixels, i.e. populations, is obtained after an down-sampling is performed, as illustrated in Methods). The parameters of the neuron model, such as the membrane potential, the threshold voltage, etc. have been set to the values reported in Table 1. The other parameters of the model (connectivity, external currents, and adaptation) are inferred from the data. This is achieved by defining the log-likelihood $\mathcal{L}$ for the parameters $\{\xi\}$ to describe a specific observed dynamics $\{S\}$ as

$$\mathcal{L}(\{S\}|\{\xi\}) = \sum_{i,t} \left[ -\frac{(S_i(t+dt) - F_i(t))^2}{2c^2} \right] \quad (1)$$

where $F_i(t)$ is the transfer function of the population $i$ and $c$ is the variance of the Gaussian distribution from which the activity is extracted (more details about this can be found in the Methods section). We remark that in this work we consider populations

composed of excitatory/inhibitory neurons. To model this, it is necessary to use an effective transfer function describing the average activity of the excitatory neurons and accounting for the effect of the inhibitory neurons (see Methods for details, *Excitatory-Inhibitory module, the effective transfer function* section).

The optimal values for the parameters are inferred from maximizing the likelihood with respect to the parameters themselves. The optimizer we used is the gradient-based iRprop algorithm[41]. The parameters are not assumed to be isotropic in the space, and each population (identified by a pixel in the image of the cortex) is left free to have different inferred parameters. The resulting parameters can thus be reported in a map, as illustrated in Fig. 3b-g. Each panel represents, color-coded, the spatial map of a different inferred parameter, describing its distribution over the cortex of the mouse. Specifically, $\lambda$, $k_0$, $e$, and $a$ characterize the inferred shape of the elliptical exponential-decaying connectivity kernels at a local level (see Methods, section Elliptic exponentially-decaying connectivity kernels); $b$ is the strength of the spike frequency adaptation (see section Generative Model and Eq. (4)); $I_{ext}$ is the local external input current. Panel **H** in Fig. 3 presents, for a few exemplary pixels, the total amount of incoming connectivity $k$, and confirms the presence of a strong eccentricity ($e$) and prevalent orientations ($\phi$) for the elliptic connectivity kernels. Both of which contribute to the initiation and propagation of wave-fronts along preferential axes, orthogonal to the major axis of the ellipses.

In principle, such inferred parameters define the generative model which reproduces best the experimental data. A major risk

while maximizing the likelihood (see Fig. 3i, blue line) is over-fitting, which we avoid by considering the validation likelihood (see Fig. 3i, orange line), i.e., the likelihood evaluated on data expressing the same kind of dynamics but not used for the inference. A training likelihood increase associated with a validation likelihood decrease during the inference procedure is an indicator for over-fitting. This is, however, not the case here.

Notably, there is no spatial regularization in the likelihood. The spatial smoothness of parameters naturally comes out when inferring the model from data.

*The outer loop.* The likelihood maximization (*inner loop*) constrains the dynamics at the following time step, given the activity at the previous step. Therefore, we expect that running a simulation with the inferred parameters should give a good prediction of the activity of the network. However, it is not straightforward that the spontaneous activity generated by the model would reproduce the desired variety of dynamical properties observed in the data. Indeed, the constraints imposed within the *inner loop* are local in time (i.e., the constraint at a certain time step $t$ only depends on the state of the network at the previous time step $t-1$). As a consequence, the model is not able to generate the long-term temporal richness observed in data. In other words, when running the spontaneous mean-field simulation for the same duration of the original recording, we found that parameters estimated in the *inner loop* can be adopted to obtain a generative model capable of reproducing oscillations and travelling wavefronts. However, the activity produced by the model inferred from the *inner loop* results is found to be much more regular as compared to the experimental activity. For instance, the down state duration and the wavefront shape express almost no variability when compared to experimental recordings (see Fig. 3, and Supp. Mat. Fig. S6).

For this reason, it is necessary to introduce the *outer loop* here described (see Fig. 4a). First, we analyze the spontaneous activity of the generative model and compare it to the data, to tune the parameters that cannot be optimized by direct likelihood differentiation (see Fig. 4a). Thus, we include an external oscillatory neuro-modulation in the model (inspired by[32]), and look for the optimal amplitude and period of the neuro-modulation proxy, expected to affect both the values of $I_{ext}$ and $b$ (more details in Methods section).

The identification of the optimal values for these parameters requires the definition of a quantitative *score* to evaluate the similarity between data and simulation. On the whole set of waves, we computed the cumulative distributions (i.e. on the whole set of waves) of the three local observables characterizing the identified travelling waves (speed, direction, and IWI, already introduced in Section Characterization of the Cortical Slow Waves, Fig. 4c, d). The distance between cumulative distributions we chose is the Earth Mover's Distance (EMD, see Methods section for more details). EMD is separately evaluated over each of these observables (Fig. 4e) in a grid search. We then combined the three EMDs as in Eq. (20). The resulting "Combined Distance" is reported in Fig. 4d for a single trial, and is further constrained by an additional requirement that excludes the zone marked in gray: we reject those simulations with too long down-states and too short up-states, if compared to experimental distributions (see purple and yellow lines in Fig. 4, respectively). Further details about the rejection criteria can be found in Methods Section The Outer Loop: grid search and data-simulation comparison.

The combination of such distances as a function of the two hyper-parameters (amplitude and period) leads to the identification of the optimal point (where the "Combined Distance" is minimal) reported as a green cross in Fig. 4d. The comparison

between experimental and simulated cumulative macroscopic distributions for a "good" (optimal match) and a "bad" (non-optimal match) point in the grid search is depicted in panels 4b and 4c, respectively. We applied the procedure to both mice, 6 recorded trials per mouse, and calibrated the model on each trial independently. Optimal values for amplitude and period for each of them are depicted in Fig. 4f and g, respectively. The distributions of the inferred neuro-modulation period (T) and amplitude (A) (Fig. 4f and g, respectively) are consistent between the two mice. The period mildly oscillates around 1 Hz. Instead, we observe more variability in the inferred amplitudes: this is reasonable, because the level of anesthesia changes among trials. In panel 4h, on the other hand, the span of combined EMD measurements for each trial in both mice is shown (within the range of values for amplitude and period considered in the grid search), depicting the variety of the observed phenomena. The black points report the "Combined Distance" for the simulation resulting from the inner loop. Moreover, Fig. 4h shows that the best "Combined Distance" achieved in the grid search (Fig. 4h, bottom of the candle) is lower than the worst one (Fig. 4h, top of the candle), and also than the one obtained for the inner-loop simulation (Fig. 4h, black points). Indeed, looking at the grid search results in Fig. 4d, the row corresponding to an amplitude $A = 0$ reports results for simulations without neuro-modulation, i.e. the inner-loop output. We observe that, according to the metrics we defined, the simulation outcome is much worse than the optimal point indicated with the green cross.

**Validation of the simulation through GMM-based propagation modes.** As shown in Fig. 5, the sequential application of the inner and the outer loop (i.e., the two-step inference model) results in an evident improvement of the simulation. Specifically, when looking at the raster plot, the neuro-modulation (outer loop) appears to be a key ingredient able to reduce the level of stereotypization of the wave collection, and to introduce elements that mimic the richness and variability of the biological phenomenon.

To further stress out this aspect, we introduce a color code for labeling waves belonging to the different clusters identified as distinct propagation modes by GMM when the model is applied to the collection of experimental and simulated waves (see Suppl. Mat. Detecting the number of components in the GMM for more details on how the algorithm detects the number of clusters). Indeed, besides illustrating the propagation features of the wave collections (as in Fig. 2), we use GMM as a tool for posterior validation of the simulation outcome (Fig. 6). The idea behind this approach is that GMM, acting on a wave collection, is able to identify distinct clusters, grouping wave events with comparable spatio-temporal features, if each identified cluster is significantly populated. This happens if the wave collection to which the GMM is applied for fitting the propagation modes is auto-consistent, i.e. contains propagation events that represent instances of the same coherent phenomenon. It is worth noting that the features fit by GMM are the spatio-temporal propagation pattern of each wave (i.e. a point in the spatially downsampled channel space). These features differ from those used by the outer loop (i.e. local velocity, direction and IWI cumulative distributions), thus providing a validation for the experiment-simulation comparison.

Results reported in Fig. 6 suggest this interpretation: here, for each mouse, the wave collection to which the GMM is applied is made up by putting together wave events from the six experimental trials and wave events from the corresponding simulated trials. In Fig. 6a and e it is reported the comparison between experimental and simulated distributions of wave

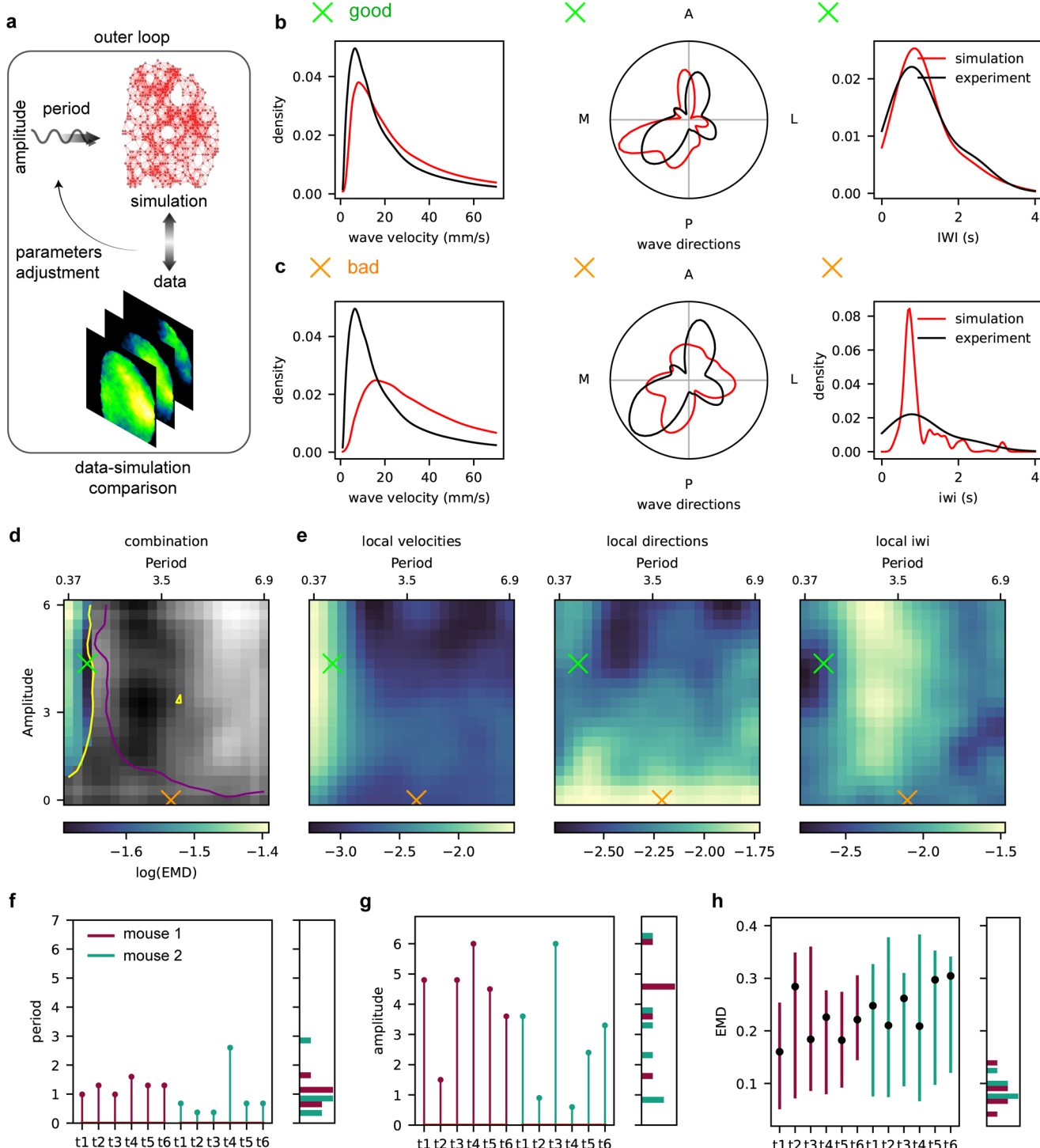

**Fig. 4 Outer Loop. a** The second step of the inference method consists in comparing simulations and data, to optimize neuro-modulation amplitude and period hyper-parameters. Here, we have a bi-directional flow of information from data to model and the other way round. **b** Direct comparison of the distributions of velocity (left panel), waves direction (central panel), and IWI (right panel) between data and simulation. The simulation is obtained selecting the optimal point (i.e. the minimum "Combined Distance" between experimental and simulated distributions) for the hyper-parameters grid search (amplitude and period of the neuro-modulation signal, see panel **d**). **c** Same as panel **b**, but for a nonoptimal point of the grid search (see panel **d**). **d** Results of the hyper-parameters grid search on a single trial. "Combined Distance" between simulated and experimental distributions (see text for details) is plotted as a function of the two hyper-parameters (amplitude and period). The gray area depicts the simulations rejected because of too long down-states (yellow line) or too short up-states (purple line), if compared with experimental distributions. The green cross depicts the optimum value, while the orange cross is a non-optimum, in fact corresponding to the inner-loop case (A = 0). **e** EMD between experimental and simulated distributions of velocities (left), directions (center) and IWI (right), for the same single trial of panel **d**. **f, g** Summary (across trials, and for the two different mice) of optimal hyper-parameter values obtained from the grid search, and their distribution (on the right). **h** Summary for the span between minimal and maximal "Combined Distances" for all the trials of the two mice, for the grid search in the same ranges of amplitude and period values of panel **d**. Black points refer to "Combined Distance" from the inner-loop simulation. On the right, the histogram of the lower bound of the EMD span is shown.

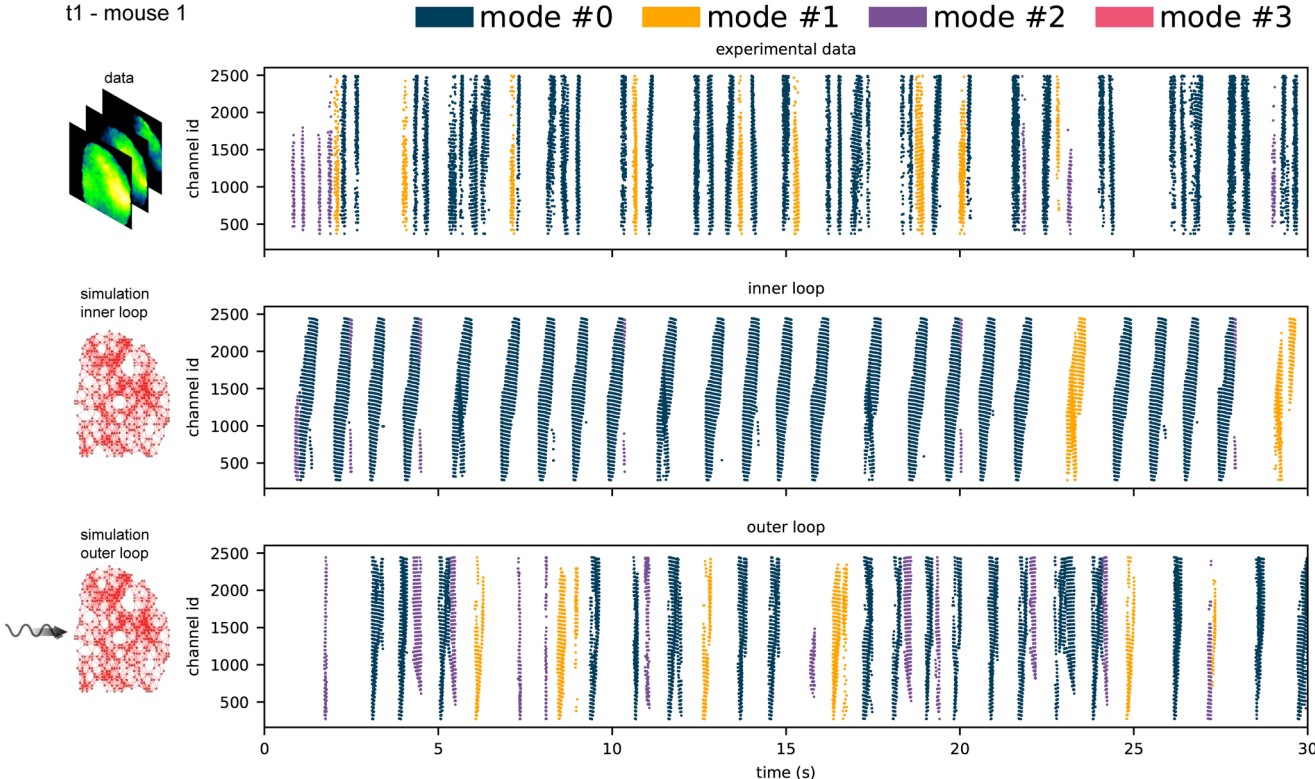

**Fig. 5 Dynamics comparison.** Raster plot of the detected waves in experimental data (top), in simulated data without neuro-modulation (output of the inner loop, center), and in simulated data with optimal neuro-modulation (output of the outer loop, bottom). For visualization purposes, only 30 s out of the 40 s of the recording (mouse 1, trial 1) are reported. All the waves detected in the three scenarios are classified though a GMM approach; the four GMM modes are identified from fitting the collection of experimental + simulated waves of all trials. In this chunk specifically, the fourth mode (pink) is not found. Wave colors indicate the resulting modes of propagation.

velocity (left panel), direction (central panel), and IWI (right panel). GMM acting on the entire collection (experiments + simulations) identifies 4 propagation modes; despite differences between the two mice, results illustrated in the pie charts (panels Fig. 6b and f) show that the GMM is "fooled" by the simulation, meaning that the clustering process on the entire collection does not trivially separate experimental from simulated events. Or rather, experimental and simulated waves are nicely mixed when events are grouped in clusters identified by the GMM, implying a correspondence between data and simulations.

A validation of this is depicted in panels Fig. 6c and g. Specifically, we compared the optimal simulation with a control case obtained by "shuffling" the output of the simulation through channel permutation equally for all the simulated waves. The GMM correctly reports a segregation between data and numerical events, suggesting that the GMM is not deceived by a simulation that is properly approaching the data. In Fig. 6c and g, it is shown the fraction of occupancy of waves identified in the shuffled dataset. This case is explored in Supplementary Note 1.

In other words, we can claim that data and simulations obtained from the two-step inference model here presented express the same propagation modes (centroids) as shown in Fig. 6d, h, while the modes expressed by the "shuffled" simulation (as a control) are orthogonal to modes found in data.

## Discussion

In this paper, we have proposed a two-step inference method to automatically build a high-resolution mean-field model of the whole cortical hemisphere of the mouse.

In recent years, the field of statistical inference applied to neuronal dynamics has mainly focused on the reliable estimate of

an effective synaptic structure of a network of neurons[42–44]. In[45] the authors demonstrated, on an in-vitro population of ON and OFF parasol ganglion cells, the ability of a Generalized Linear Model (GLM) to accurately reproduce the dynamics of the network[46]; in addition, they studied the response properties of lateral intra-parietal area neurons at the single-trial, single-cell level. The capability of GLM to capture a broad range of single neuron response behaviors was analyzed in[47]. However, in these works, the focus was on the response of neurons to stimuli of different spatio-temporal complexity. Moreover, recurrent connections, even when accounted for, did not provide a decisive correction to the network dynamics. To date, a very few published studies have focused on the spontaneous dynamics of networks inferred from neuronal data. Most of them focused on the resting state recorded through fMRI and on the reproduction of its functional connectivity[22]. In[23] the temporal and spatial features of bursts propagation have been accurately reproduced, but on a simpler biological system, a culture of cortical neurons recorded with a 60 electrodes array. Here we aim at reproducing the dynamics of the whole hemisphere of the mouse with approximately 1400 recording sites.

One of the major risks when inferring a model is the failure of the obtained generative model to reproduce a dynamics comparable to the data. The reason for this is the difficulty to constrain some of the observables when inferring a model. A common example is offered by the Ising model: when inferring its couplings and external fields, only magnetizations and correlations between different spins are constrained[24], but it is not possible to constrain the power spectrum displayed by the model and other temporal features. When this experiment is done on the Ising model itself, this is not a problem: if the dataset is large

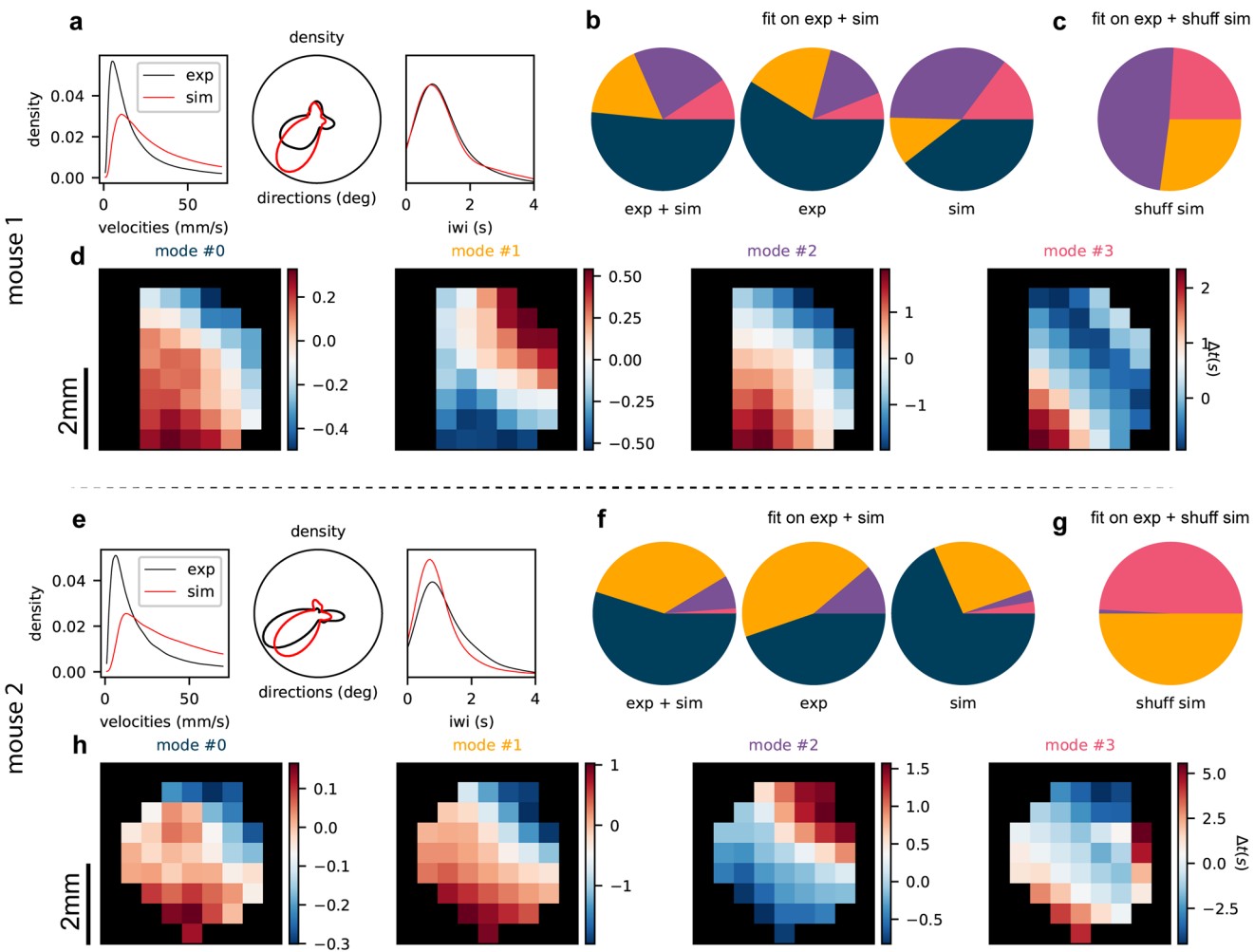

**Fig. 6 Validation of propagation modes.** Analysis over the full set of 6 experimental trials and corresponding optimal simulations (for each of the two mice). Optimal simulations are selected trial-wise following the grid search approach, as described in section The Outer Loop. **a** and **e** Direct comparison between experimental and simulated distributions of wave velocity (left panel), direction (central panel), and IWI (right panel). **b** and **f** Fraction of each of the detected modes in the full collection (experiments + simulations, left), in the sub-collection of experiments only (center), and simulations only (right). **c** and **g** Modes fraction of shuffled simulated waves classified by a GMM fit over shuffled simulation + experimental dataset (see Fig. S3 in Supplementary Note 1 for a detailed description). Shuffled simulation: a posteriori validation of the simulated model. **d** and **h**. Centroids of the 4 gaussian distributions (modes) in the downsampled channel space describing the collection of experimental and simulated waves fitted with the GMM.

enough, the correct parameters are inferred and also the temporal features are correctly reproduced. However, if the original model contains some hidden variables (unobserved neurons, higher-order connectivities, other dynamical variables), the dynamics generated by the inferred model might not be representative of the original dynamics. This led us to introduce the two-step method, in order to adjust parameters beyond likelihood maximization.

Another obvious but persisting problem in the application of inference models to neuroscience has been how to assess the value of inferred parameters, as well as their biological interpretation. In the literature, a cautionary remark is usually included, recognizing that inferred couplings are to be meant as "effective", leaving of course open the problem of what exactly this means.

One way to infer meaningful parameters is to introduce good priors. In this work, we assumed that under anesthesia the inter-areal connectivity is largely suppressed in favor of lateral connectivity, as demonstrated e.g. in[28] during a similar oscillatory regime. Also, the probability of lateral connectivity in the cortex is experimentally known[27] to decay with the distance according to an exponential shape, with a spatial scale in the range of a few

hundreds of μm. Furthermore, the observation of preferential directions for the propagation of cortical slow waves[48] suggested us to explicitly account for such anisotropy. For all of these reasons, we introduced exponentially decaying elliptical connectivity kernels in our model. We remark that inferring connections at this mesoscopic scale allows obtaining information complementary to that contained in inter-areal connectivity atlases, where connectivity is considered on a larger spatial scale, and it is carried by white matter rather than by lateral connectivity.

Another possible approach to get priors could be to make use of Structural Connectivity (SC) atlases that, however, ignore individual variations in anatomic structures and are usually based on ex-vivo samples for which the structure can be modified by brain tissue processing[49–51]. Also, SC atlases fail to track the dynamic organization of functional cortical modules in different cognitive processes[52,53]. Indeed, a key feature of neural networks is *degeneracy*, the ability of systems that are structurally different to perform the same function[49]. In other words, the same task can be achieved by different individuals with different neural architectures.

On the other hand, Functional Connectivity (FC) is individual-specific an activity-based measure still highly dependent on physiological parameters like brain state and arousal so that FC atlases might not be representative of the subject under investigation[54]. E.g., state-dependent control of circuit dynamics by neuromodulatory influences including acetylcholine (ACh) strongly affects cortical FC[55]. In addition, functional modules often vary across individual specimens and different studies[52,56,57].

In summary, generic atlases of Structural Connectivity (SC) and Functional Connectivity (FC) contain information averaged among different individuals. Though similar functions and dynamics can be sustained in each individual by referring to specific SCs and FCs[50,51], detailed information about individual brain peculiarities are lost with these approaches. In the framework here presented, detailed individual connectivities can be directly inferred in vivo, therefore complementing the information coming from SC and FC, and contributing to the construction of brain-specific models descriptive of the dynamical features observed in one specific individual.

One of the values added by the modeling technique here described is that its dynamics relies on a mean-field theory, which describes populations of interconnected spiking neurons. This is particularly useful in order to set up a richer and more realistic dynamics in large-scale spiking simulations of slow waves and awake-like activity[58], that in absence of such detailed inferred parameters behave according to quite stereotyped spatio-temporal features.

In the framework of the activity we are carrying out in the Human Brain Project, aiming at the development of a flexible and modular analysis pipeline for the study of cortical activity[35] (see also[59]), we developed for this study a set of analysis tools applicable to both experimental and simulated data, capable to identify wavefronts and quantify their features.

In perspective, we plan to integrate plastic cortical modules capable of incremental learning and sleep on small-scale networks with the methodology here described, demonstrating the beneficial effects of sleep on cognition[38,60,61].

This paper is based on data acquired under a Ketamine-Xylazine anesthesia. An example of application with potential therapeutical impact is to contribute to the understanding of the mechanisms underlying the effect of ketamine treatments in therapy of depression[62].

Generally speaking, there are strong evidences of oscillatory neuro-modulation currents, affecting the cortex and coming from deeper brain areas (such as the brain-stem), with a sub-Hertz frequency[63,64]. However, we acknowledge that this shape for the neuro-modulation might have a limited descriptive power, and we plan to develop more general models. A possible choice can be that of a combination of many oscillatory modes or noisy processes such as Ornstein-Uhlenbeck and gaussian processes. However, the risk of this kind of models is overparametrization. This raises the necessity of a penalty to discourage a too large number of parameters and avoid over fitting (see e.g. Akaike Information Criterion[65]).

Finally, we stress once again that in the current work we only considered spontaneous dynamics. Then, we plan to go beyond spontaneous activity, aiming at modeling the dynamics of slow waves when the network receives input signals[66,67], ranging from simple (local) electrical perturbations to proper sensory stimuli. The methodology to achieve this would be very similar to what presented in[68–70]. We present an example of the response of the inferred network to a pulse stimulation in Supp. Mat. Section Simulation with pulse stimulation.

## Methods

**Mouse wide-field calcium imaging.** In-vivo calcium imaging datasets have been acquired by LENS personnel (European Laboratory for Non-Linear Spectroscopy (LENS Home Page, http://www.lens.unifi.it/index.php)) in the Biophotonics laboratory of the Physics and Astronomy Department, Florence, Italy. All the procedures were performed in accordance with the rules of the Italian Minister of Health (Protocol Number 183/2016-PR). Mice were housed in clear plastic enriched cages under a 12 h light/dark cycle and were given *ad libitum* access to water and food. The transgenic mouse line C57BL/6J-Tg(Thy1GCaMP6f)GP5.17Dkim/J (referred to as GCaMP6f mice (For more details, see The Jackson Laboratory, Thy1-GCaMP6f, https://www.jax.org/strain/025393; RRID:IMSR_JAX:025393)) from Jackson Laboratories (Bar Harbor, Maine, USA) was used. This mouse model selectively expresses the ultra-sensitive calcium indicator (GCaMP6f) in excitatory neurons.

Surgery procedures and imaging: 6-month-old male mice were anaesthetized with a mix of Ketamine (100 mg/Kg) and Xylazine (10 mg/Kg). The procedure for implantation of a chronic transcranial window is described in[16,17]. Briefly, to obtain optical access to the cerebral mantle below the intact skull, the skin and the periosteum over the skull were removed following application of the local anesthetic lidocaine (20 mg/mL). Imaging sessions were performed right after the surgical procedure. Wide-field fluorescence imaging of the right hemisphere were performed with a 505 nm LED light (M505L3 Thorlabs, New Jersey, United States) deflected by a dichroic mirror (DC FF 495-DI02 Semrock, Rochester, New York USA) on the objective (2.5x EC Plan Neofluar, NA 0.085, Carl Zeiss Microscopy, Oberkochen, Germany). The fluorescence signal was selected by a band-pass filter (525/50 Semrock, Rochester, New York USA) and collected on the sensor of a high-speed complementary metal-oxide semiconductor (CMOS) camera (Orca Flash 4.0 Hamamatsu Photonics, NJ, USA). A 3D motorized platform (M-229 for the $xy$ plane, M-126 for the $z$-axis movement; Physik Instrumente, Karlsruhe, Germany) allowed the correct subject displacement in the optical setup.

**Experimental data and deconvolution.** We assumed that the optical signal $X_i(t)$ acquired from each pixel $i$ is proportional to the local average excitatory (somatic and dendritic) activity (see Fig. 1). The main issue is the slow (if compared to the spiking dynamics) response of the calcium indicator (GCaMP6f). We estimate the shape of such a response function from the single-spike response[7] as a log-normal function:

$$\text{LN}\left(\frac{t}{dt}; \mu, \sigma\right) = \frac{dt}{t} \frac{1}{\sqrt{2\pi}\sigma} \exp\left(-\frac{\left(\ln\frac{t}{dt} - \mu\right)^2}{2\sigma^2}\right), \quad (2)$$

where $dt = 40$ ms is the temporal sampling frequency, and $\mu = 2.2$ and $\sigma = 0.91$ have been estimated in[16]. We assumed a linear response of the calcium indicator (which is known to be not exactly true[7]) and we applied the deconvolution to obtain an estimation of the actual firing rate time course $S_i(t)$. The deconvolution is achieved by dividing the signal by the log-normal in the Fourier space:

$$\hat{S}_i(k) = \frac{\hat{X}_i(k)}{\hat{LN}(k)} \Theta(k_0 - k) \quad (3)$$

where $\hat{X}_i(k) = fft(X_i(t))$ and $\hat{LN}(k) = fft(LN(t))$ (fft( · ) is the fast Fourier transform). Finally, the deconvolved signal is $S_i(t) = ifft(\hat{S}_i(k))$ (ifft( · ) is the inverse fast Fourier transform). The function $\Theta(k_0 - k)$ is the Heaviside function and is used to apply a low-pass filter with a cut-off frequency $k_0 = 6.25$ Hz.

**Generative Model.** We assumed a system composed of $N_{pop}$ (approximately 1400, as the number of pixel) interacting populations of neurons. The population $j$ at time $t$ has a level of activity (firing rate) defined by the variable $S_j(t)$ (expressed in ms$^{-1}$). As anticipated, each population $j$ is associated to a pixel $j$ of the experimental optical acquisition, and contains $N_j$ neurons; $J_{ij}$ and $C_{ij}$ are the average synaptic weight and the average degree of connectivity between populations $i$ and $j$, respectively.

We defined the parameter $k_{ij} = N_j J_{ij} C_{ij}$ as the relevant one in the inference procedure. We also considered that each population receives an input from a virtual external population composed of $N_j^{ext}$ neurons, each of which injecting a current $J_j^{ext}$ in the population with a frequency $\nu_j^{ext}$. Similarly to what done above, we defined the parameter $I_j^{ext} = \nu_j^{ext} N_j^{ext} J_j^{ext}$.

In a common formalism, site $i$ at time $t$ is influenced by the activity of other sites through couplings $k_{ik}$[23], in terms of a local input current defined as

$$\mu_i(t) = \sum_k k_{ik} S_k(t) + \frac{I_i^{ext}}{C_m} - \frac{b_i W_i(t)}{C_m} \quad (4)$$

The term $-\frac{b_i W_i(t)}{C_m}$ accounts for a feature that plays an essential role in modeling the emergence of cortical slow oscillations: the spike frequency state-dependent adaptation, that is the progressive tendency of neurons to reduce their firing rates even in the presence of constant excitatory input currents. Spike frequency adaptation is known to be much more pronounced in deep sleep and anesthesia than during wakefulness. From a physiological stand-point, this is associated to the calcium concentration in neurons and can be modeled introducing a negative (hyper-polarizing) current on population $i$ (the $- b_i W_i(t)$ term in Eq. (4)).

$W(t)$ is the adaptation (adimensional) variable: it increases when a neuron $i$ emits a spike and is gradually restored to its rest value when no spikes are emitted;

its dynamics can be written as

$$W_i(t+dt) = (1-\alpha_w)W_i(t) + \alpha_w \tau_w S_i(t+dt) \tag{5}$$

where $\alpha_w = 1 - \exp(-dt/\tau_w)$, and $\tau_w$ is the characteristic time scale of spike frequency adaptation.

It is also customary to introduce the adaptation strength $b_i$ that determines how much the fatigue influences the dynamics: this factor changes depending on neuro-modulation and explains the emergence of specific dynamic regimes associated to different brain states. Specifically, higher values of $b$ (together with higher values of recurrent cortico-cortical connectivity) are used in models for the induction of Slow Oscillations.

It is possible to write down the dynamics for the activity variables by assuming that the probability at time $t + dt$ to have a certain level of activity $S_i(t+dt)$ in the population $i$ is drawn from a Gaussian distribution:

$$p(S_i(t+dt)|\mathbf{S}(t)) = p(S_i(t+dt)|F_i(t)) = \frac{1}{\sqrt{2\pi c^2}} \exp\left[-\frac{(S_i(t+dt)-F_i(t))^2}{2c^2}\right]. \tag{6}$$

For simplicity, we assumed $c = 2\,\mathrm{s}^{-1}$ to be constant in time, and the same for all the populations. Its temporal dynamics might be accounted for by considering a second-order mean-field theory[36,71].

The average activity $S_i(t+dt)$ of the population $i$ at time $t+dt$ is computed as a function of its input by using a leaky-integrate-and-fire response function

$$\langle S_i(t+dt) \rangle = F[\mu_i(t), \sigma_i^2(t)] := F_i(t) \tag{7}$$

where $F[\mu_i(t), \sigma_i^2(t)]$ is the transfer function that, for an AdEx neuron under stationary conditions, can be analytically expressed as the flux of realizations (i.e., the probability current) crossing the threshold $V_{spike} = \theta + 5\Delta V$[40,72]:

$$F(\mu) = F(\mu, \sigma^2 = \sigma_\star^2) = \frac{1}{\sigma_\star^2} \int_{-\infty}^{\theta+5\Delta V} dV \int_{\max(V,V_r)}^{\theta+5\Delta V} du \exp\left\{-\frac{1}{\tau_m \sigma_\star^2} \int_V^u [f(v) + \mu\tau_m] dv\right\}. \tag{8}$$

where $f(v) = -(v(t) - E_l) + \Delta V \exp\{(v(t)-\theta)/\Delta V\}$, $\tau_m$ is the membrane time constant, $E_l$ is the reversal potential, $\theta$ is the threshold, $C_m$ is the membrane capacitance, and $\Delta V$ is the exponential slope parameter. The parameter values are reported in Table 1. Here we considered a small and constant $\sigma_\star^2 = 10^{-4}\,\mathrm{mV^2/s}$, since we assumed small fluctuations, and fluctuations have a small effect on the population dynamics in any case. We will verify such assumption a posteriori. It is actually straightforward to work out the same theory without making any assumption on the size of $\sigma$, the only drawback being an increased computational power required to evaluate a 2D function (in Eq. (8)) during the inference (and simulation) procedure.

In a first-order mean-field formulation, the gain function depends on the infinitesimal mean $\mu$ and variance $\sigma^2$ of the noisy input signal. $\mu$ is proportional to $K = NJp$ (number of neurons, weights, and connectivity, respectively). However, $Jp$ can be rescaled in order to keep $K$ fixed. On the other hand, $\sigma^2$ is proportional to $K^2/(Np)$, thus necessarily depending on $N$. However, this latter aspect is neglected in our current formulation (e.g. see Eq. (8)), as we assumed a constant small value for the infinitesimal variance, allowing to arbitrarily choose $N$, with the proper rescaling. In order to be compliant with neuronal densities coming from physiological literature (about 45 K neurons per mm$^2$) we took $N = 500$ in our model.

**The inner loop: likelihood maximization**. It is possible to write the log-likelihood of one specific history $\{S_i(t)\}$ of the dynamics of the model, given the parameters $\{\xi_k\}$, as follows:

$$\mathcal{L}(\{S\}|\{\xi\}) = \sum_{i,t} \left[-\frac{(S_i(t+dt)-F_i(t))^2}{2c^2}\right] \tag{9}$$

The optimal parameters, given a history of the system $\{S(t)\}$, can be obtained by maximizing this log-likelihood. When using a gradient-based optimizer (we indeed used iR-prop), it is necessary to explicitly compute the derivatives

$$\frac{\partial \mathcal{L}(\{S\}|\{\xi\})}{\partial \xi_k} = -\sum_{i,t} \left[\frac{(S_i(t+dt)-F_i(t))}{c^2}\right] \frac{\partial F_i(t)}{\partial \mu_i} \frac{\partial \mu_i}{\partial \xi_k} \tag{10}$$

where $\frac{\partial F_i(t)}{\partial \mu_i}$ are computed numerically, while $\frac{\partial \mu_i}{\partial \xi_k}$ can be easily evaluated analytically. The value of $c^2$ is not relevant since it can be absorbed in the learning rate.

The optimization procedure was performed independently on the 12 data chunks of 40 s here considered. For each chunk, parameters were actually optimized on the first 32 s, while the remaining 8 s were used for validation. In more detail, the likelihood evaluated on such a validation dataset with the values of the parameters inferred on former datasets is referred to as validation likelihood, and helps to prevent the risk of overfitting.

**Excitatory - inhibitory module, the effective transfer function**. In this paper, the single population (pixel) was assumed to be composed of two sub-populations of excitatory and inhibitory neurons; the mean of the input currents reads as:

$$\begin{cases} \mu_i^e(t) = \sum_k k_{ik}^{ee} S_k^e(t) + \sum_k k_{ik}^{ei} S_k^i(t) + I_{\text{ext}}^e - b_i W_i(t) \\ \mu_i^i(t) = \sum_k k_{ik}^{ie} S_k^e(t) + \sum_k k_{ik}^{ii} S_k^i(t) + I_{\text{ext}}^i \end{cases} \tag{11}$$

It is not always possible to distinguish the excitatory $S^e$ from the inhibitory firing rates $S^i$ in experimental recordings. Indeed, in electrical recordings, the signal is a composition of the two activities. However, in our case we can make the assumption that the recorded activity, after the deconvolution, is a good proxy of the excitatory activity $S^e$. This is possible thanks to the fluorescent indicator we considered in this work, which targets specifically excitatory neurons. In this way, it is possible to constrain the excitatory activity and to consider $S^i$ as hidden variable.

The transfer function depends both on the excitatory and the inhibitory activities $F_e(S^e, S^i)$. However, $S^i$ is not observable in our dataset, and it is necessary to estimate an effective transfer function for the excitatory population of neurons by making an adiabatic approximation on the inhibitory one[73]. We defined the effective transfer function as

$$F(S^e) = F_e^{\text{eff}}(S^e) = F_e(S^e, S^i(S^e)) \tag{12}$$

The inhibitory activity $S^i$ is assumed to have a faster time scale. In first approximation, a biologically plausible assumption when comparing the activity of fast-spiking inhibitory neurons with regular spiking excitatory neurons. As a consequence, $S^i(S^e)$ can be evaluated as its stationary value given a fixed $S^e$ value. The likelihood then contains this new effective transfer function as a function of $k^{ee}$ and $k^{ei}$, which are the weights to be optimized. The dependence on $k^{ie}, k^{ii}$ disappears: they have to be suitably chosen (we set $k^{ie} = k^{ii} = -25\,\mathrm{mV}$ in our case) and cannot be optimized with this approach.

**Elliptic exponentially decaying connectivity kernels**. Having a large number of parameters, especially in the case of small datasets, can cause convergence problems and can increase the risk of overfitting. However, experimental constraints and priors can be used to reduce the number of parameters to be inferred. In our case, we decided to exploit the information that lateral connectivity decays with the distance, in first approximation according to a simple exponential reduction law[27], with long-range inter-areal connection suppressed during the expression of slow waves[28]. Furthermore, we supposed that a deformation from a circular symmetry to a more elliptical shape of the probability of connection could facilitate the creation of preferential directions of propagation observed in the experiments.

Therefore, we included this in the model as an exponential decay on the spatial scale $\lambda$. In this case the average input current to the population $i$ can be written as already presented in Eq. (4), with $k_{ik}$ taking into account the above-discussed assumptions:

$$\mu_i(t) = \sum_k k_{ik} S_k(t) + \frac{I_{ext}}{C_m} - \frac{b_i W_i(t)}{C_m}, \qquad k_{ij} = k_0^j \exp\left(-\frac{d_{ij}}{\lambda_j}\right) \tag{13}$$

where $d_{ik}$ is the distance between populations (i.e. pixels) $i$ and $k$,

$$d_{ik} = \rho_{ik}\left(1 + e_k \cos(2\theta_{ik} + 2\phi_k)\right)\left(1 + a_k \cos(\theta_{ik} + \phi_k)\right) \tag{14}$$

Parameters can be inferred by differentiating such parametrization of the connectivity kernel. For example, the derivative with respect to the spatial scale $\lambda_k$ to feed Eq. (10) would be:

$$\frac{\partial \mu_i(t)}{\partial \lambda_k} = k_{ik} S_k(t) \frac{d_{ik}}{\lambda_k^2} \tag{15}$$

**Gaussian mixture model**. Collected the transition times to the up state in each of the $L$ downsampled channels (44 and 41 informative channels for the two mice, respectively), each wave is hence described by a vector $\mathbf{x}$ in a $L$-dimensional space. The problem of classifying slow waves into typical spatio-temporal patterns of propagation can then be tackled as a clustering problem in such high-dimensional space. We chose the approach based on Gaussian Mixture Model (GMM)[74].

We initially assumed $K$ typical propagation patterns, each described by a $L$-dimensional multi-variate Gaussian, with its mean $\boldsymbol{\mu}_k$ and covariance matrix $\boldsymbol{\Sigma}_k$. The probability of a wave $\mathbf{x}_n$ to belong to the $k$-th cathegory is hence given by the related multi-variate Gaussian density function:

$$p(\mathbf{x}_n|k) = \mathcal{N}(\mathbf{x}_n; \boldsymbol{\mu}_k, \boldsymbol{\Sigma}_k) \tag{16}$$

Ignoring which cluster each wave belongs to, one has to sum over all the possibilities, taking into account the relative weights $\boldsymbol{\pi}$ of the $K$ Gaussians in the mixture. The total likelihood for the set $\mathbf{X}$ of waves hence reads:

$$p(\mathbf{X}) = \prod_{n=1}^N p(\mathbf{x}_n) = \prod_{n=1}^N \sum_{k=1}^K \pi_k \mathcal{N}(\mathbf{x}_n; \boldsymbol{\mu}_k, \boldsymbol{\Sigma}_k) \tag{17}$$

in fact depending on the set of Gaussian means and covariances $\theta \equiv \{\boldsymbol{\mu}_k, \boldsymbol{\Sigma}_k\}$, $p(\mathbf{X}) = p(\mathbf{X}; \theta)$, as well as on the number $K$ and the relative proportions $\boldsymbol{\pi}$ of propagating modes.

The scope of this clustering procedure is in first place to infer the features of typical propagation patterns, i. e. their mean $\boldsymbol{\mu}$ and their covariance $\boldsymbol{\Sigma}$. Then, each wave is assigned to one of these modes. To this aim, maximum-likelihood approaches are exploited, pointing at finding the optimal values of parameters $\theta$

that maximize the likelihood $\mathcal{L}(\theta) \equiv p(X; \theta)$ from Eq. (17). This can be easily attained through the Python library *sklearn.mixture.GaussianMixture*[33], exploiting the Expectation-Maximization (EM) algorithm for likelihood maximization.

Notice that EM is guaranteed to converge to a local maximum of the likelihood. This implies that, for non-convex problems, the solution to which the algorithm converges depends on the initial condition. To get rid of this dependence, a possible strategy is to run the maximization by reshuffling the data and finally averaging over the final configurations of clusters.

In fact, the number of typical propagation patterns $K$ is not even known a priori, so it has to be inferred as well. To face this problem, we run the above procedure for different numbers $K$ of clusters and compare the resulting performances, exploiting a Gaussianity test (*scipy.stats.normaltest*[75]) as a quantitative parameter for comparison.

**Neuro-modulation**. The periodic neuro-modulation we included in our network was modeled as a periodic oscillation of the parameters $b$ and $I^{\text{ext}}$ described by the following equations:

$$\begin{cases} I_i^{\text{ext}}(t) = I_i^{\text{ext},0}\left(1 + A\cos\left(\frac{2\pi t}{T}\right)\right) \\ b_i(t) = b_i^0\left(1 + \frac{A}{2}\cos\left(\frac{2\pi t}{T}\right)\right) \end{cases} \tag{18}$$

where we defined $I_i^{\text{ext},0}$ and $b_i^0$ as the set of parameters inferred in the inner loop, while $A$ and $T$ are the amplitude and the oscillation period optimized in the outer loop.

**The outer loop: grid search and data-simulation comparison**. There can be different ways to define the outer loop. One way might be an iterative algorithm where at each step simulation and data are compared, and parameters are updated with some criterion.

Here we used a simpler strategy, implementing a grid search of the parameters to be optimized, in order to find the best match between simulations and data. The parameters we considered are $A$ and $T$, namely the amplitude and the period of the neuro-modulation (Eq. (18)). We run a 21(T) × 22(A) grid search with parameter $A$ linearly ranging from 0.0 to 6, and parameter $T$ linearly ranging from 0.37 s to 6.9 s. This choice was performed heuristically. However, it is possible to run a wider and denser grid search at the cost of a higher computational expense.

Thus, we processed the simulation run over for each couple of values $(A, T)$ in the grid, applying the same pipeline of analysis applied to experimental data, obtaining in output the distribution of the three previously defined local observables: wave speed, propagation direction and inter-wave interval (see Section Slow Waves and Propagation analysis for more details).

We quantitatively compared each of these distributions with the ones observed in experimental data applying an Earth Mover's Distance (EMD) measure. Specifically, given two distributions $u$ and $v$, EMD can be seen as the minimum amount of work (i.e. distribution weight that needs to be moved multiplied by the distance it has to be moved) required to transform $u$ into $v$. In a more formal definition, EMD is also known as the Wasserstein distance between the two 1D distributions, namely

$$\text{EMD}(u, v) = \inf_{\pi \in \Gamma(u,v)} \int_{\mathbb{R}\times\mathbb{R}} |x - y| \, d\pi(x, y) = \int_{-\infty}^{+\infty} |U(s) - V(s)| \, ds \tag{19}$$

where $\Gamma(u, v)$ is the set of (probability) distributions on $\mathbb{R}\times\mathbb{R}$ whose marginals are $u$ and $v$ on the first and second factors, respectively. $U$ and $V$ are the respective cumulative density functions (CDFs) of $u$ and $v$. The proof of the equivalence of both definitions can be found in[76].

Specifically, for the purposes of this work, we evaluated the CDFs of each distribution sampling with a discrete binning. Moreover, in order to have this measure independent from the characteristic bin scale of the three studied observables, we evaluated EMD for speed, directions and IWI in "bin units": thanks to this approach, we can combine these measurements without incurring into non-homogeneous measure unit inconsistencies.

Then, to define a quantitative "score" describing the similarity between data and simulation, we chose to combine the three EMDs computed over the three macroscopic local observables in an Euclidean way

$$\text{EMD} = \sqrt{\text{EMD}_{\text{speed}}^2 + \text{EMD}_{\text{direction}}^2 + \text{EMD}_{\text{IWI}}^2} \tag{20}$$

Finally, we imposed two constraints on the simulation output, in order to reject biologically meaningless values for $A$ and $T$: 1) most of the down-state duration distribution and 2) most of the wave duration distribution need to be both lower than their minimum values observed in experimental data (after having removed outliers; acceptance criteria: 98th and 95th percentile, respectively). This allowed us to exclude an entire region of non-interesting simulations and identify the optimal, biologically meaningful results.

**Slow waves and propagation analysis**. To compare the simulation outputs with experimental data we implemented and applied the same analysis tools to both sets. Specifically, we improved the pipeline described by some of us in[16,35] to make it more versatile.

First, experimental data were arranged to make them suitable for both the application of the slow wave propagation analysis and the inference method. Thus, data were loaded and converted into a standardized format; the region of interest was selected by masking channels of low-signal intensity; fluctuations were further reduced applying a spatial down-sampling: pixels were assembled in 2 × 2 blocks through the function *scipy.signal.decimate*[75], curating the aliasing effect with the application of an ideal Hamming filter. Then, data were ready to be both analyzed or given as input to the inference method.

Both experimental data and simulation output can then be given as input to these analysis tools. The procedure we followed is articulated into four blocks:

- Processing: the constant background signal is estimated for each pixel as the mean signal computed on the whole image set; it is then subtracted from images, pixel by pixel. The resulting signal is then normalized by its maximum.
- Trigger Detection: transition times from Down to Up states are detected. These are identified from the signal local minima. Specifically, the collection of transition times for each channel is obtained as the vertex of the parabolic fit of the minima.
- Wave Detection: the detection method used in[16] and described in[77] is applied. It consists in splitting the set of transition times into separate waves according to a *Unicity Principle* (i.e. each pixel can be involved in each wave only once) and a *Globality Principle* (i.e. each wave needs to involve at least 75% of the total pixel number).
- Wave Characterization: once that the wave collection has been identified, such set of waves is characterized by measuring local wave velocities, local wave directions and local SO frequencies.

Specifically, we computed the local wave velocity as

$$v(x, y) = \frac{1}{|\nabla T(x, t)|} = \frac{1}{\sqrt{\left(\frac{\partial T(x,y)}{\partial x}\right)^2 + \left(\frac{\partial T(x,y)}{\partial y}\right)^2}} \tag{21}$$

where $T(x, y)$ is the passage time function of each wave. Having access to this function only in the pixel discrete domain, the partial derivatives have been calculated as finite differences

$$\begin{cases} \frac{\partial T(x,y)}{\partial x} \simeq \frac{T(x+d,y) - T(x-d,y)}{2d} \\ \frac{\partial T(x,y)}{\partial y} \simeq \frac{T(x,y+d) - T(x,y-d)}{2d} \end{cases}.$$

where $d$ is the distance between pixels.

Concerning local directions, for each wave transition we computed a weighted average of wave local velocity components through a Gaussian filter $w_{(\mu, \sigma)}$ centered in the pixel involved in the transition and with $\sigma = 2$. The local direction associated to the transition is

$$\theta(x, y) = \tan^{-1}\left(\frac{\langle v_y \rangle_{w_{(y,\sigma)}}}{\langle v_x \rangle_{w_{(x,\sigma)}}}\right) \tag{22}$$

Finally, the SO local frequency $f_{\text{SO}}$ is computed as the inverse time lag of the transition on the same pixel by two consecutive waves. Defining $T^{(w)}(x, y)$ as the transition time of wave $w$ on pixel $(x, y)$, the SO frequency can thus be computed as

$$f_{\text{SO}}^{(1,2)}(x, y) = \frac{1}{T^{(2)}(x, y) - T^{(1)}(x, y)} \tag{23}$$

**Statistics and reproducibility**. The model is built upon several recordings coming from two different mice (six for each mouse). The obtained results are remarkably comparable between the two mice, indicating a good reproducibility of our method.

We performed validation on our model and acquired long simulations in order to conduct a grid search for optimal parameters. This ensured the accuracy and reliability of our results.

## Data availability
Experimental wide-field calcium imaging recordings of anesthetized mice are publicly available datasets from the EBRAINS Knowledge Graph platform at the link https://kg.ebrains.eu/search/instances/Dataset/28e65cf1-ce13-4c12-92dc-743b0cb66862[78].

## Code availability
The implementation of the inference method and data-analysis is available on GitHub at the link https://github.com/APE-group/CorticalSW_Inference, together with the source codes to reproduce the figures in this paper.

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

## Acknowledgements
This work has been supported by the European Union Horizon 2020 Research and Innovation program under the FET Flagship Human Brain Project (grant agreement SGA3 n. 945539 and grant agreement SGA2 n. 785907) and by the INFN APE Parallel/Distributed Computing laboratory.

## Author contributions
C.C. contributed to conceptualization, methodology, software, investigation, data curation, formal analysis, supervision, validation, visualization; C.D.L. contributed to conceptualization, methodology, software, investigation, data curation, formal analysis, supervision, validation, visualization; G.D.B. contributed to conceptualization, methodology, investigation, data curation; R.G. contributed to methodology, software, computational resources; I.B. contributed to methodology; E.P. contributed to methodology, validation, visualization; F.S. contributed to computational resources; C.L. contributed to methodology, visualization; L.T. contributed to methodology; F.R. contributed to experiments, investigation, data curation; A.L.A.M. contributed to experiments, investigation, data curation; F.P. contributed to experiments, investigation, data curation; M.D. contributed to methodology, computational resources; P.S.P. contributed to conceptualization, methodology, investigation, project administration, supervision, funding acquisition. All authors contributed to writing and revising the manuscript.

## Competing interests
The authors declare no competing interests.
