## [Peer Review File · Communications Biology]

Reviewers' comments:

Reviewer #1 (Remarks to the Author):

Revision "Simulations Approaching Data: Cortical Slow Waves in Inferred Models of the Whole Hemisphere of Mouse"

The work uses the high-resolution in-vivo brain recordings of 2 anaesthetized mice obtained by wide-field calcium imaging techniques. The authors propose a 2-step inference procedure compound by an inner loop that optimizes by likelihood maximization a mean-field model, and the outer loop that optimizes the period and amplitude of a periodic neuro-modulation to best fit observables from the empirical wave propagation. Overall, I found the manuscript to be well-organized and the results are informative for a broad set of literature and suggest interesting new ideas for the field to test. Nevertheless, I found some points that could be important that the authors clarify in order to strengthen the paper.

Major:

- The motivation is not completely clear. For instance, the authors claim that motivation is more methodological (using a computational model to describe the complexity in brain dynamics). In my opinion, the paper's main question should be clearer and interestingly based on a neurobiological question. This is reflected in the discussion that seems to lose integration in the last paragraph and it reads like a taxonomy of possible interpretations or results of the work. It would be great if the authors could clarify and focus on a question that are able to be answered with their proposal (e.g., describe the anaesthesia condition by the pie plots of modes distribution and relate each mode with some neurobiological feature).

-The introduction should be more clear and direct; there are paragraph that reads more like methods

- The author said that: "the sequential application of the inner and the outer loop (i.e., the two-step inference model) results in an evident improvement of the simulation."

In my opinion, would be great if this sentence could be supported. It is not surprising that if you are adding two new parameters over the inference step that are able to not only increase the number of parameters that you are optimizing but also the dynamical repertoire of your system, the results of the fitting will improve.

The optimization procedure is split into the inner and outer loops, but: could be done in one step only? What happens if you use an optimization algorithm to find the optimal parameter that best fits the EMD regarding 8 parameters at the same time (the 6 of the inner and the 2 of the outer loop)? It would be great if the authors explained more about the advantages of the 2 steps procedure over 1 step procedure alternative. Or even better, why is this two-step procedure necessary for the optimization?

- In the analysis presented in Fig. 6 the authors claim that : "we can claim that data and simulations obtained from the two-step inference model here presented to express the same propagation modes".

In that case they perform the GMM to an extended data set including empirical data + simulated data and obtain a set of waves modes from this data set, and when they create the extended data with shuffled simulated data, the results are different. This null test is not reflected in Fig. It would be great if despite showing the pie from only simulated and only simulation, in this figure author show the comparison between emp+sim and emp+sim shuffled (like Fig 7, where is not clear the difference between both situations).

On the other hand, in my opinion, a comparison to be done in order to confirm what the authors are claiming is to train the GMM only with experimental data, define the clusters (the modes), and assess how behaves the simulated data when is forced to the empirical modes. Have you obtained a similar pie of modes distribution in that case between empirical and simulated?

Minor:

- the order of the references. References should be numerically ascending in order of appearance.
- There is twice the phrase that you propose a two-step method at the beginning of the section and in the last paragraph of page 3.

Fig 1.

- Please rewrite clearer the caption for panel B; what is referring to "left", and where is the transfer function?
- Change panel D the rows to be consistent with the order of simulation and data (top simulation; bottom data).

Fig2:

- There is no error in the wave velocity and IWI between mice across trials? Are you displaying the mean? One trial? It seems like these plots are schemes for both measures and they seem equally for both mice. The y-axis values are missing. Please clarify it.
- "the defined experimental observables are remarkably comparable among the two subjects since they display similar distributions in detected waves velocity, directions and IWI."◇ to quantify how similar they are could help to understand also the variability across trials (in line with my previous comment of fig2)
- The spatial maps in figure 3 seem to be quite smooth. Do you have any constraint of continuity in the space when you inferred the parameters in the inner loop?
- last paragraph page 7, Is describing the outer loop but referring to Fig 3, but should be Fig 4.
- The first sentence of the discussion says that you are working with rat brains and the rest of the ms is about mice.

Reviewer #2 (Remarks to the Author):

This paper demonstrates a method for fitting a spatially-varying mean-field model to calcium imaging data exhibiting slow waves. The fitting method involves two stages, each sensitive to different statistics of the wave data. The model appears to fit quite well, and the two-step fitting method is interesting and may be useful for other groups to apply to their own models. The modelling doesn't reveal any deep biological insights here, but in my opinion the fitting method is interesting enough on its own to carry the paper. Given the growing interest in wave analyses this paper would likely find a good audience. My comments focus more on clarifications. The paper would be improved if the authors address the following:

1. It is clear that the two-step fitting process yields a good fit to the data. The two steps are performed sequentially, the former fitting most of the model parameters (fixing the two

neuromodulation parameters) and the outer loop then fits the remaining two parameters. If the full model were written with the neuromodulation (Eq. 18) dynamics included, could a single-step fit using the "inner loop" likelihood maximization method work to fit the whole model? Is the problem that A and T are assumed uniform across the grid rather than having a subscript i ? Given the well-behaved spatial plots in Fig. 3B-G of the inferred local parameters, it seems that even a spatially-varying A and T would be tractable here.

2. Overall I found the writing not as clear as would be ideal. For such a methods-focused paper, the "results first" format means there are a lot of forward references to the methods and supporting information. The paper would be clearer if the methods came before the results; if journal style prohibits this, then it would be clearer if more of the model and inversion method text was integrated into the results. After all, the model and fitting method more or less ARE the main results of the paper. I also think the supporting information is sufficiently short that it can be integrated into the paper (at worst as an appendix), particularly since the results section cites equations down there.

3. "Following this route, a small number of models have been able to reproduce the complexity of observed dynamics from a single recording. However, most of the models are focused on the fMRI resting state recordings and on the reproduction of the functional connectivity [2]." -- This is an overly narrow view. Mean field models have been fitted to EEG spectra (e.g. van Albada et al. 2010 Clin Neurophys, and numerous other works from Robinson et al.), DCM has been used in various flavors of many papers including for fitting to EEG, and models have been fitted to seizures in SEEG data (e.g. the Virtual Epileptic Patient, Jirsa et al. 2017 NeuroImage) and ECoG data (e.g. Karoly et al. 2018 PLOS CB).

4. abstract, "The model is capable to reproduce most of the features of the non-stationary and non-linear dynamics displayed", it is unclear where nonstationary and nonlinear dynamics were addressed in the paper.

5. The authors note that they "exploited a minimal amount of generic anatomical knowledge", namely exponentially decaying lateral connectivity (this is well-established) and that long-range connectivity is suppressed during waves. Is this latter phenomenon a case of suppressed `_structural_` connectivity (what would that mean physically?) or suppressed `_functional_` connectivity? It is unclear that the latter would be an anatomical prior on connectivity versus something that should emerge in the model dynamics.

6. Fig. 2D, the velocities seem lower than the mode/median/mean of the distributions in Fig. 2A, can the authors elaborate on why?

7. Can the authors elaborate on how the GMM used here identifies modes? From Fig. 5 it appears that waves are relatively discrete and non-overlapping in time. Are the modes necessarily distinct in time or can they be superposed?

8. It is stated that each neural mass is composed of 500 AdEx neurons, but are these individual neurons used for anything or only the masses? If it's only the masses it seems that the initial number of neurons is not relevant. Clarify.

9. In Fig. 3I it's a bit odd to put the legend over so much of the panel. And what's the y axis scale? Does it matter?

10. In the outer loop section, references to Fig. 3A should I think be 4A.

11. Fig. 4E is not referenced in the text.

12. Can the authors interpret the results shown in Fig. 4D-H? It's not clear what the reader is meant

to take away here.

13. Fig 5 top vs bottom, it's interesting that the frequency of the appearance of waves is lower than in the data, I would have thought that the fitting of the neuromodulation period might have captured this slower time scale. Any comment on the discrepancy?

14. Fig. 6C and F pie charts, I like the idea of throwing the model and data in together to show that they are not trivially distinguished and hence comparable, but can the authors make it clearer how to see this 'the GMM is "fooled"' point? The three pie charts all appear to be fairly different.

15. Are the equations dimensionally correct? It appears that $f(v)$ is a voltage, then in Eq. 8, $\mu \cdot \tau$ has the same dimensions as $f(v)$ but is the product of a current and a time constant, so is a charge rather than a voltage.

16. " $k_{ie} = k_{ii} = -25$ in our case" -- units?

17. The authors varied the number of clusters used in the GMM, it would be helpful to see how their performance measure depended on the number of clusters.

18. "parameter T linearly ranging from 1.2s to 2s." -- Fig. 4 suggests a higher upper bound was used. And is period T hitting the upper limit in most of the epochs?

19. Labeling is too small in many of the figures.

20. There should be a space between a number and its units, and units shouldn't be italic.

21. p17 last line, "2x2" -- \times not x.

Reviewer #3 (Remarks to the Author):

This paper presents an intriguing data- and model-driven analysis of travelling waves of cortical activity in high-resolution in-vivo recordings of mouse brain obtained from wide-field calcium imaging. The work integrates cutting edge data analysis methods for wave detection with a novel "two loop" technique for model and data assimilation.

The paper is very well written and clearly presented. It will definitely be of substantial interest to experimentalists using high field imaging techniques as well as researchers using computational models. I have a few suggestions for the authors consider in their revisions:

Major:

1. The model fitting looks interesting, yields plausible dynamics and is well motivated. But in terms of (perhaps future work), only one sort of dynamic is possible (periodically modulated non stationary GMM modes): The authors should outline next steps to allow different generative models to be posed and a means of identifying the best of the competing models of the data (also incorporating a complexity penalty as otherwise the most complicated model will likely win).

2. The sanity check on page 8 + figure 6 is very nice but does seem a bit circular in the sense that the same data and model are used to generate the simulated data, then test/validate the mix of experimental and simulated data. Ideally, the model should be trained and tested on different data (for cross-validation) and also the performance of the method in detecting the mix should use a metric which differs from that used to optimise the model fit.

3. A related test would be to mix up experimental and simulated data where the latter are generated in a manner that is a poor fit to the experiment – then use the approach to ensure the 2 (experimental and counter-factual simulations) are not mixed by the model inference technique.

4. Since the authors adopt an exponential synaptic kernel, I suggest they link their approach to neural field theory, which derives a PDF based on a very similar form (see e.g. Robinson, et al (1997)).

Propagation and stability of waves of electrical activity in the cerebral cortex. *Physical Review E*, 56(1), 826

5. I'm not convinced of the strictly periodic nature of neuromodulation (although it likely is adequate for the present purposes). Most likely neuromodulation is a heavy-tailed stochastic process, or even a weakly damped mean reverting process. Some discussion of other choices for the form of equation (18) should be considered

Minor:

1. The paper is in general very clearly written. However, the abstract is a bit idiosyncratic. I suggest rewording the abstract to something like:

"The development of novel techniques to record wide-field brain activity enables estimation of data-driven models from thousands of recording channels and hence across large regions of cortex. These in turn improve our understanding of the modulation of brain-states and the richness of traveling waves dynamics. Here, we infer data-driven models from high-resolution in-vivo recordings of mouse brain obtained from wide-field calcium imaging. We then assimilate experimental and simulated data through the characterization of the spatio-temporal features of cortical waves in experimental recording. Inference is built in two steps: an inner loop that optimizes a mean-field model by likelihood maximization, and an outer loop that optimizes a periodic neuro-modulation via direct comparison of observables that characterize cortical slow waves. The model reproduces most of the features of the non-stationary and non-linear dynamics present in the high-resolution in-vivo recording of the mouse brain. The proposed approach offers new methods of characterizing and understanding cortical waves for experimental and computational neuroscientists.

2. "However, the activity produced by the model inferred from the inner loop results is found to be much more regular as compared to the experimental activity. For instance, the down state duration and the wavefront shape express almost no variability when compared to experimental recordings." – as evidenced by what features in which figure/panel?

3. The text at the bottom of p7 should refer to Fig 4 (not Fig 3).

4. I think the Discussion should briefly canvass some of the prior empirical work on cortical travelling waves in experimental data: see some suggested references below (no need to cite all of them!!)

Muller, L., Reynaud, A., Chavane, F. & Destexhe, A. The stimulus-evoked population response in visual cortex of awake monkey is a propagating wave. *Nat. Commun.* 5, 3675 (2014).

Rubino, D., Robbins, K. A. & Hatsopoulos, N. G. Propagating waves mediate information transfer in the motor cortex. *Nat. Neurosci.* 9, 1549–1557 (2006).

Muller, L. & Destexhe, A. Propagating waves in thalamus, cortex and the thalamocortical system: experiments and models. *J. Physiol. Paris* 106, 222–238 (2012).

Townsend, R. G. et al. Emergence of complex wave patterns in primate cerebral cortex. *J. Neurosci.* 35, 4657–4662 (2015).

Hindriks, R., van Putten, M. J. A. M. & Deco, G. Intra-cortical propagation of EEG alpha oscillations. *Neuroimage* 103, 444–453 (2014).

Burkitt, G. R., Silberstein, R. B., Cadusch, P. J. & Wood, A. W. Steady-state visual evoked potentials and travelling waves. *Clin. Neurophysiol.* 111, 246–258 (2000).

Muller, L., Chavane, F., Reynolds, J. & Sejnowski, T. J. Cortical travelling waves: mechanisms and computational principles. *Nat. Rev. Neurosci.* 19, 255–268 (2018).

Sato, T. K., Nauhaus, I. & Carandini, M. Traveling waves in visual cortex. *Neuron* 75, 218–229 (2012).

Prechtl, J. C., Cohen, L. B., Pesaran, B., Mitra, P. P. & Kleinfeld, D. Visual stimuli induce waves of electrical activity in turtle cortex. *Proc. Natl Acad. Sci. USA* 94, 7621–7626 (1997).

Besserve, M., Lowe, S. C., Logothetis, N. K., Schölkopf, B. & Panzeri, S. Shifts of gamma phase across primary visual cortical sites reflect dynamic stimulus- modulated information transfer. *PLoS Biol.* 13, e1002257 (2015).

Townsend, R. G., Solomon, S. S., Martin, P. R., Solomon, S. G. & Gong, P. Visual motion discrimination by propagating patterns in primate cerebral cortex. *J. Neurosci.* 37, 10074–10084 (2017).

Benucci, A., Frazor, R. A. & Carandini, M. Standing waves and traveling waves distinguish two circuits in visual cortex. *Neuron* 55, 103–117 (2007).

Rebuttal letter - Revision 1

Simulations Approaching Data: Cortical Slow Waves in Inferred Models of the Whole Hemisphere of Mouse

Reviewer #1 (Remarks to the Author):

The work uses the high-resolution in-vivo brain recordings of 2 anaesthetized mice obtained by wide-field calcium imaging techniques. The authors propose a 2-step inference procedure compound by an inner loop that optimizes by likelihood maximization a mean-field model, and the outer loop that optimizes the period and amplitude of a periodic neuro-modulation to best fit observables from the empirical wave propagation. Overall, I found the manuscript to be well-organized and the results are informative for a broad set of literature and suggest interesting new ideas for the field to test. Nevertheless, I found some points that could be important that the authors clarify in order to strengthen the paper.

Major:

R1.1 The motivation is not completely clear. For instance, the authors claim that motivation is more methodological (using a computational model to describe the complexity in brain dynamics). In my opinion, the paper's main question should be clearer and interestingly based on a neurobiological question. This is reflected in the discussion that seems to lose integration in the last paragraph and it reads like a taxonomy of possible interpretations or results of the work. It would be great if the authors could clarify and focus on a question that are able to be answered with their proposal (e.g., describe the anaesthesia condition by the pie plots of modes distribution and relate each mode with some neurobiological feature).

Reviewer 3 kindly suggested a reformulation of the abstract that highlights the scientific question. Furthermore, we added in the introduction and discussion the following text:

Introduction, line **96**:

<<In summary, this paper addresses the understanding of the mechanisms underlying the emergence of the spatio-temporal features of cortical waves leveraging the integration of two aspects: the knowledge coming from experimental data and the interpretation gained from simulations. We identified essential ingredients needed to reproduce the main modes expressed by the biological system, providing a mechanistic explanation grounded in the neuro-modulation and spatial heterogeneity of connectivity and local parameters.>>

Discussion, line **377**:

<<This paper is based on data acquired under a Ketamine-Xylazine anesthesia. An example of application with potential therapeutical impact is to contribute to the understanding of the mechanisms underlying the effect of ketamine treatments in therapy of depression [Oliver et al. 2022].>>

[Oliver 2022] Oliver, P. A., Snyder, A. D., Feinn, R., Malov, S., McDiarmid, G., & Arias, A. J. (2022). Clinical effectiveness of intravenous racemic ketamine infusions in a large community sample of patients with treatment-resistant depression, suicidal ideation, and

generalized anxiety symptoms: a retrospective chart review. *The Journal of Clinical Psychiatry*, 83(6), 42811.

R1.2 The introduction should be more clear and direct; there are paragraph that reads more like methods.

We reformulated the introduction to be more clear and less methodological.

R1.3a The author said that: “the sequential application of the inner and the outer loop (i.e., the two-step inference model) results in an evident improvement of the simulation.” In my opinion, would be great if this sentence could be supported. It is not surprising that if you are adding two new parameters over the inference step that are able to not only increase the number of parameters that you are optimizing but also the dynamical repertoire of your system, the results of the fitting will improve.

We included the following considerations in the main text at line **264**:

<<Indeed, looking at the grid search results in Fig. 4D, the row corresponding to an amplitude $A = 0$ reports results for simulations without neuro-modulation, i.e. the inner-loop output. We observe that, according to the metrics we defined, the simulation outcome is much worse than the optimal point indicated with the green cross. >>

R1.3b The optimization procedure is split into the inner and outer loops, but: could be done in one step only? What happens if you use an optimization algorithm to find the optimal parameter that best fits the EMD regarding 8 parameters at the same time (the 6 of the inner and the 2 of the outer loop)? It would be great if the authors explained more about the advantages of the 2 steps procedure over 1 step procedure alternative. Or even better, why is this two-step procedure necessary for the optimization?

To answer this question, we included an additional section in the supplementary material: ***Inferring neuromodulation parameters through likelihood maximization***

R1.4 In the analysis presented in Fig. 6 the authors claim that : “we can claim that data and simulations obtained from the two-step inference model here presented express the same propagation modes”.

In that case they perform the GMM to an extended data set including empirical data + simulated data and obtain a set of waves modes from this data set, and when they create the extended data with shuffled simulated data, the results are different. This null test is not reflected in Fig. It would be great if despite showing the pie from only simulated and only simulation, in this figure author show the comparison between emp+sim and emp+sim shuffled (like Fig 7, where is not clear the difference between both situations). On the other hand, in my opinion, a comparison to be done in order to confirm what the authors are claiming is to train the GMM only with experimental data, define the clusters (the modes), and assess how behaves the simulated data when is forced to the empirical modes. Have you obtained a similar pie of modes distribution in that case between empirical and simulated?

Fig 7 (in the old version, now Fig S3, sup. mat.) is meant to be an extension of Figure 6: indeed, panels B,C,E and F of Fig. 6 are exactly the same panels shown in old Fig 7 B,F,I and M respectively. In order to make it more clear and emphasize the a-posteriori validation of the model, we moved the “A posteriori validation of the simulated wave modes” section from supplementary material to the main text and re-organized the old Fig 6 and 7 (now Fig 6 and **Fig S3** in supplementary material).

We also thank the reviewer for suggesting an additional validation measure. We split the experimental wave collection in a training- and a testing dataset, fit the GMM only on the training one (three modes are thus identified) and measured how well the testing experimental dataset, the simulated one and the shuffled collection are represented by the identified modes, with agreement expressed in terms of log-likelihood.

These results are summarized in the text **lines 46**, in supplementary material.

<<As a first validation analysis of the fidelity of the model, we forced the optimal simulation and the control (shuffled) simulation over the empirical (experiment-only fitted) modes. The fraction of detected modes is represented in Fig.S3D, K. We also measured how well these modes describe the simulated datasets. To do so, we split the experimental dataset in a training and a testing dataset, we fit the GMM over the training one only (75% of waves) and measured the log-likelihood associated with the classification of the testing experimental dataset (upper bound), the simulated waves datasets and the shuffled control dataset (lower 52 bound) as depicted in Table 1. The confidence in the representation of the simulated dataset is $88 \pm 2\%$ (mouse 1) and $85 \pm 1\%$ (mouse 2).>>

Figure 6: Validation of propagation modes. Analysis over the full set of 6 experimental trials and corresponding optimal simulations (for each of the two mice). Optimal simulations are selected trial-wise following the grid search approach, as described in section The Outer Loop. A and E. Direct comparison between experimental and simulated distributions of wave velocity (left panel), direction (central panel), and IWI (right panel). B and F. Fraction of each of the detected modes in the full collection (experiments + simulations, left), in the sub-collection of experiments only (center), and simulations only (right). C and G. Modes fraction of shuffled simulated waves classified by a GMM fit over shuffled simulation + experimental dataset (see Fig. S3 in Supplementary Material for a detailed description). Shuffled simulation: a posteriori validation of the simulated model. D and H. Centroids of the 4 gaussian distributions (modes) in the downsampled channel space describing the collection of experimental and simulated waves fitted with the GMM.

Figure S3: A posteriori validation of simulated wave modes, for the two mice separately, comparing the optimal simulation with a control case (shuffled simulation). A, G. Wave propagation modes identified by GMM fitted on experimental waves only (3 identified modes). D, L. Fraction of each identified mode when forced to classify experimental dataset (left), simulated dataset (center) and shuffle simulated dataset (right) on the empirical modes. B, H. Wave propagation modes identified by GMM fitted on both experimental and simulated waves (4 identified modes). E, M. Fraction of each of the detected modes in the collection of both experimental and simulated waves (left), and in the sub-collections of experimental waves (center) and simulated waves (right). C, I. Wave propagation modes identified by GMM fitted on experimental and shuffled simulated waves (4 identified modes). N. Fractions of identified modes in the collection of both experimental and shuffled simulated waves (left), and in the sub-collections of experimental waves (center) and shuffled simulated waves (right). All the waves are shuffled through the same permutation of channels.

Minor:

R1.5

the order of the references. References should be numerically ascending in order of appearance.

Bibliography order has been sorted according to appearance.

R1.6a There is twice the phrase that you propose a two-step method at the beginning of the section and in the last paragraph of page 3.

We thank the reviewer to have us noticed this oversight. We have corrected line 125 accordingly.

R1.6b

Fig 1.

-Please rewrite clearer the caption for panel B; what is referring to “left”, and where is the transfer function?

The caption of panel B in Figure 1 has been updated and elements not present in the figure have been removed.

R1.7

Fig 1.

-Change panel D the rows to be consistent with the order of simulation and data (top simulation; bottom data).

We might have misunderstood the question here. Panel D in Figure 1 has simulation in the top and experimental data in the bottom, accordingly with the schematic representation of inner and outer loop in panel C. We kindly ask the reviewer to provide more detail about this request.

R1.8 Fig2:

- There is no error in the wave velocity and IWI between mice across trials? Are you displaying the mean? One trial? It seems like these plots are schemes for both measures and they seem equally for both mice. The y-axis values are missing. Please clarify it.

Figure 2 represents the cumulative distribution of local velocities, directions and IWIs across all trials for each mouse, i.e. histograms of values obtained at each pixel considering, for each subject, the full set of waves across the full set of trials. Figure 4, on the other hand, depicts the experimental and simulated distributions of those macroscopic observables only for a single trial in mouse 1.

To better show the variability among trials we included an additional section in the Supplementary Material (**Variability across trials**) and added Fig.S1.

Figure S1: Summary of wave propagation properties in experimental trials for the two analysed mice. Measures for the quantitative characterization and comparison of waves cumulated across all pixels in each trial in mouse 1 (left) and 2 (right), respectively. A-B. Cumulative distributions of local wave velocities, directions and inter-wave intervals for each trial and each mouse. C-D. EMD between the macroscopic distributions of the same observable from different trials of the same mouse, for each of the two mice.

R1.9 Fig2:

- “the defined experimental observables are remarkably comparable among the two subjects since they display similar distributions in detected waves velocity, directions and IWI.” to quantify how similar they are could help to understand also the variability across trials (in line with my previous comment of fig2)

See answer to R1.8, specifically, we quantitatively measured the distance between the macroscopic characterization of the observed phenomena in different trials computing the earth moving distance between such distributions (panels C and D of **Figure S1**)

R1.10 Fig3:

The spatial maps in figure 3 seem to be quite smooth. Do you have any constraint of continuity in the space when you inferred the parameters in the inner loop?

We added the following text at line **216**.

<<Notably, there is no spatial regularization in the likelihood. The spatial smoothness of parameters naturally comes out when inferring the model from data.>>

R1.11

last paragraph page 7, Is describing the outer loop but referring to Fig 3, but should be Fig 4.

Yes, definitely. Thank you very much for having noticed this mistake. We have corrected lines **232, 234** accordingly.

R1.12

The first sentence of the discussion says that you are working with rat brains and the rest of the ms is about mice.

We thank the reviewer for noticing this oversight. We have corrected line **306** accordingly.

Reviewer #2 (Remarks to the Author):

This paper demonstrates a method for fitting a spatially-varying mean-field model to calcium imaging data exhibiting slow waves. The fitting method involves two stages, each sensitive to different statistics of the wave data. The model appears to fit quite well, and the two-step fitting method is interesting and may be useful for other groups to apply to their own models. The modelling doesn't reveal any deep biological insights here, but in my opinion the fitting method is interesting enough on its own to carry the paper. Given the growing interest in wave analyses this paper would likely find a good audience. My comments focus more on clarifications. The paper would be improved if the authors address the following:

In order to highlight the non-methodological aspects of the work, we added the following text in the introduction and discussion:

Introduction, line **96**:

<<In summary, this paper addresses the understanding of the mechanisms underlying the emergence of the spatio-temporal features of cortical waves leveraging the integration of two aspects: the knowledge coming from experimental data and the interpretation gained from simulations. We identified essential ingredients needed to reproduce the main modes expressed by the biological system, providing a mechanistic explanation grounded in the neuro-modulation and spatial heterogeneity of connectivity and local parameters.>>

Discussion, line **377**:

<<This paper is based on data acquired under a Ketamine-Xylazine anesthesia. An example of application with potential therapeutical impact is to contribute to the understanding of the mechanisms underlying the effect of ketamine treatments in therapy of depression [Oliver et al. 2022].>>

[Oliver 2022] Oliver, P. A., Snyder, A. D., Feinn, R., Malov, S., McDiarmid, G., & Arias, A. J. (2022). Clinical effectiveness of intravenous racemic ketamine infusions in a large community sample of patients with treatment-resistant depression, suicidal ideation, and generalized anxiety symptoms: a retrospective chart review. *The Journal of Clinical Psychiatry*, 83(6), 42811.

R2.1. It is clear that the two-step fitting process yields a good fit to the data. The two steps are performed sequentially, the former fitting most of the model parameters (fixing the two neuromodulation parameters) and the outer loop then fits the remaining two parameters. If the full model were written with the neuromodulation (Eq. 18) dynamics included, could a single-step fit using the "inner loop" likelihood maximization method work to fit the whole model? Is the problem that A and T are assumed uniform across the grid rather than having a subscript i ? Given the well-behaved spatial plots in Fig. 3B-G of the inferred local parameters, it seems that even a spatially-varying A and T would be tractable here.

We included the following considerations in the main text at line **264**::

<<Indeed, looking at the grid search results in Fig. 4D, the row corresponding to an amplitude $A = 0$ reports results for simulations without neuro-modulation, i.e. the inner-loop output. We observe that, according to the metrics we defined, the simulation outcome is much worse than the optimal point indicated with the green cross.>>

Also, we included an additional section in the supplementary material:

Inferring neuromodulation parameters through likelihood maximization

R2.2. Overall I found the writing not as clear as would be ideal. For such a methods-focused paper, the "results first" format means there are a lot of forward references to the methods and supporting information. The paper would be clearer if the methods came before the results; if journal style prohibits this, then it would be clearer if more of the model and inversion method text was integrated into the results. After all, the model and fitting method more or less ARE the main results of the paper. I also think the supporting information is sufficiently short that it can be integrated into the paper (at worst as an appendix), particularly since the results section cites equations down there.

We hope to have addressed all the issues raised by the reviewers, with the only exception of this R2.2 point, that suggests a reordering according to the introduction, methods results, discussion template. We ask the editor for an indication about this point.

R2.3. "Following this route, a small number of models have been able to reproduce the complexity of observed dynamics from a single recording. However, most of the models are focused on the fMRI resting state recordings and on the reproduction of the functional connectivity [2]." -- This is an overly narrow view. Mean field models have been fitted to EEG spectra (e.g. van Albada et al. 2010 Clin Neurophys, and numerous other works from Robinson et al.), DCM has been used in various flavors of many papers including for fitting to EEG, and models have been fitted to seizures in SEEG data (e.g. the Virtual Epileptic Patient, Jirsa et al. 2017 NeuroImage) and ECoG data (e.g. Karoly et al. 2018 PLOS CB).

We thank the reviewers for the interesting references, we included them in the paper by introducing the following text at line **51**:

<<In [Van Albada et al.] the authors constrain the model dynamics to reproduce the experimental spectrum. The work presented in [Jirsa et al.] proposes a network of modules called epileptors, and infers local excitability parameters. Similarly, the authors in [Karoly et al.] estimate parameters and states from a large dataset of epileptic seizures.>>

[Van Albada et al.] SJ Van Albada et al. "Neurophysiological changes with age probed by inverse modeling of EEG spectra". In: Clinical neurophysiology 121.1 (2010), pp. 21–38.

[Jirsa et al.] Viktor K Jirsa et al. "The virtual epileptic patient: individualized whole-brain models of epilepsy spread". In: *Neuroimage* 145 (2017), pp. 377–388.

[Karoly et al.] Philippa J Karoly et al. "Seizure pathways: A model-based investigation". In: *PLoS computational biology* 14.10 (2018), e1006403.

R2.4. abstract, "The model is capable to reproduce most of the features of the non-stationary and non-linear dynamics displayed", it is unclear where nonstationary and nonlinear dynamics were addressed in the paper.

Regarding the non-stationarity, we added the following text in the main text, line **75**:

<<Specifically, we demonstrate that the inclusion of a time-dependent acetylcholinic neuro-modulation term in the model enables a better match between experimental recordings and simulations, inducing a variability in the expressed dynamics otherwise stereotyped.>>

And a sections "**Non-stationary dynamics**" in the supplementary material (line **92**) with the following text and a novel figure S6.

<<The non-stationarity observed in the data, and reproduced in the model thanks to the neuro-modulation, can be easily visualized in the spectrograms reported in panel A of Fig. S6. While appreciable in both experimental and simulated (inner+outer loop) data, such non-stationarity is not observed in the spectrum generated by the inner loop solely. For a quantitative assessment, we computed the temporal auto-correlation of each frequency in the three different scenarios. In panel B, we show the normalized mean auto-correlation for the first 10s for experimental data (left), the output of the inner loop (centre), and the output of the outer loop (right). The correlation rapidly decreases in time for both experimental data and simulation, whereas it remains high for the output of the inner loop, showing a stationary behavior. Moreover, these observations are shown to be consistent both inter-trial and inter-individual.>>

The non-linearity of the dynamic is testified by the abrupt transition between up and down state observed in the data (see Fig.1B) and can be accounted for by the non-linear shape of the transfer function (eq.8) derived from the mean field theory [Capone et al.].

[Capone et al.] Cristiano Capone et al. "Slow waves in cortical slices: how spontaneous activity is shaped by laminar structure". In: *Cerebral cortex* 29.1 (2019), pp. 319–335

Figure S6: Non-stationary dynamics. A. Spectrograms for the average activity in a single trial from mouse 1, computed on experimental data (top) and simulation outcome (middle, inner loop only; bottom, outer loop). B. Normalized mean temporal auto-correlation of frequencies for both mice in experimental data (left) and simulated data (center, inner loop only; right, outer loop).

R2.5. The authors note that they "exploited a minimal amount of generic anatomical knowledge", namely exponentially decaying lateral connectivity (this is well-established) and that long-range connectivity is suppressed during waves. Is this latter phenomenon a case of suppressed `_structural_` connectivity (what would that mean physically?) or suppressed `_functional_` connectivity? It is unclear that the latter would be an anatomical prior on connectivity versus something that should emerge in the model dynamics.

To address this, we included the following text at line 82:

<<Following this route, a minimal anatomical assumption, is to decompose structural connectivities into a short-range lateral connectivity contribution (the exponential decay characterising the lateral connectivity [34], or intra-area connectivity) and a long-range white matter mediated connectivity. It is well known, and confirmed in our experimental data, that during deep sleep and deep and intermediate anesthesia, propagation is by slow wave propagation and therefore can be mainly mediated by lateral cortical connectivity ([35]). [...]. Here, we propose a methodology to implement an inverse approach, from wave properties to model parameters. The choice of local connectivity kernels reduces the number of parameters to be inferred from N^2 to N (number of recording sites). The proposed approach prevents overfitting and keeps the computational cost of inference under control even for higher resolution data.>>

R2.6. Fig. 2D, the velocities seem lower than the mode/median/mean of the distributions in Fig. 2A, can the authors elaborate on why?

As quantitatively shown in [Gutzen et al.] the distributions of wave characteristics change as a function of the downsampling factor. With a decreasing spatial resolution, fewer waves are detected, and they appear more planar as some complex local patterns are no longer

detected. In Fig.2 A, over the original dataset and the one downsampled to be fitted with the GMM. Here, the distributions are sharpened and have a lower mean, coherent with the global velocities of the typical waves identified. We added this additional measure in Fig2 and commented this result at **line 169**.

<<It is also worth noting that, as quantitatively shown in [Gutzen et al.], the distributions of wave quantitative observables change as a function of the downsampling factor. With a decreasing spatial resolution, fewer waves are detected, and they appear more planar as some complex local patterns are no longer detected. In Fig. 2A, we show the distribution of local velocities measured over the original dataset at a spatial resolution of 0.1 mm (black) and the one downsampled at a spatial resolution of 0.6 mm (orange). Here, the distributions are sharpened and have a lower mean, coherent with the global velocities of the typical waves identified. >>

[Gutzen et al.] Robin Gutzen et al. Comparing apples to apples – Using a modular and adaptable analysis pipeline to compare slow cerebral rhythms across heterogeneous datasets. 2022. doi: 10.48550/ARXIV.2211.08527. url: <https://arxiv.org/abs/2211.08527>.

R2.7. Can the authors elaborate on how the GMM used here identifies modes? From Fig. 5 it appears that waves are relatively discrete and non-overlapping in time. Are the modes necessarily distinct in time or can they be superposed?

GMM are fitted on a dataset composed by the observed spatio-temporal wave patterns represented in a spatially subsampled channel space (i.e. for each entry in the dataset, the number of features is equal to the number of reduced channels). In other words, once a wavefront is reconstructed from the spatio-temporal distribution of down-to-up transitions (as illustrated in Fig 5), it is mapped into the reduced channel space, and it constitutes a single instance in the fitting dataset.

To identify travelling waves from the raw transitions, we used the COBRAWAP pipeline [Gutzen et al.]. The procedure used is summarised in section “Slow Waves and Propagation analysis” in the Supplementary Material. To make it clearer, we added the following text in **line 163**

<<Specifically, GMM are fitted on a dataset composed by the observed spatio-temporal wave patterns represented in the subsampled channel space (i.e. for each entry in the dataset, the number of features is equal to the number of reduced channels). See section Methods, Slow Waves and Propagation analysis for a more detailed description of how travelling waves are identified from the raw signal.>>

We also moved to the main text (line 659) the paragraph “Slow Waves and Propagation analysis” that in the submission was presented in Supplementary Material

[Gutzen et al.] Robin Gutzen et al. Comparing apples to apples – Using a modular and adaptable analysis pipeline to compare slow cerebral rhythms across heterogeneous datasets. 2022. doi: 10.48550/ARXIV.2211.08527. url: <https://arxiv.org/abs/2211.08527>.

R2.8. It is stated that each neural mass is composed of 500 AdEx neurons, but are these individual neurons used for anything or only the masses? If it's only the masses it seems that the initial number of neurons is not relevant. Clarify.

To clarify this point, we included the following text at **line 470**:

<<In a first-order mean-field formulation, the gain function depends on the infinitesimal mean μ and variance σ^2 of the noisy input signal. μ is proportional to $K = N J_p$ (number of neurons, weights, and connectivity, respectively). However, J_p can be rescaled in order to keep K fixed. On the other hand, σ^2 is proportional to $K^2 / (N p)$, thus necessarily depending on N . However, this latter aspect is neglected in our current formulation (e.g. see Eq. (8)), as we assumed a constant small value for the infinitesimal variance, allowing to arbitrarily choose N , with the proper rescaling. In order to be compliant with neuronal densities coming from physiological literature (about 45 K neurons per mm^2) we took $N = 500$ in our model.>>

R2.9. In Fig. 3I it's a bit odd to put the legend over so much of the panel. And what's the y axis scale? Does it matter?

We fixed the figure.

R2.10.

In the outer loop section, references to Fig. 3A should I think be 4A.

Yes, definitely. Thank you very much for having noticed this mistake. We have corrected lines **232 and 234** accordingly.

R2.11.

Fig. 4E is not referenced in the text.

We have added some text to describe **Fig 4.E** at line **242**. The added text follows:

<<D is separately evaluated over each of these observables (Fig. 4E) in a grid search. We then combined the three EMDs as in Eq. (20).>>

R2.12. Can the authors interpret the results shown in Fig. 4D-H? It's not clear what the reader is meant to take away here.

We inserted the following text at line **255**:

<<The distributions of the inferred neuro-modulation period (T) and amplitude (A) (Figs. 4F and G, respectively) are consistent between the two mice. The period mildly oscillates around 1 Hz. Instead, we observe more variability in the inferred amplitudes: this is reasonable, because the level of anesthesia changes among trials. In panel 4H, on the other hand, the span of combined EMD measurements for each trial in both mice is shown (within the range of values for amplitude and period considered in the grid search), depicting the variety of the observed phenomena. The black points report the "Combined Distance" for the simulation resulting from the inner loop. Moreover, Fig. 4H shows that the best "Combined Distance" achieved in the grid search (Fig. 4H, bottom of the candle) is lower than the worst one (Fig. 4H, top of the candle), and also than the one obtained for the inner-loop simulation (Fig. 4H, black points).>>

Also we added the following text at line **218**

<<On the whole set of waves, we computed the cumulative distributions (i.e. on the whole set of waves) of the three local observables characterizing the identified travelling waves (speed, direction, and IWI, already introduced in Section Characterization of the Cortical

Slow Waves). The distance between cumulative distributions we chose is the Earth Mover's Distance (EMD, see Methods section for more details). EMD is separately evaluated over each of these observables (Fig. 4E) in a grid search. We then combined the three EMDs as in Eq. (20). The resulting "Combined Distance" is reported in Fig. 4D for a single trial, and is further constrained by an additional requirement that excludes the zone marked in gray: we reject those simulations with too long down-states and too short up-states, if compared to experimental distributions (see purple and yellow lines in Fig. 4, respectively).>>

R2.13. Fig 5 top vs bottom, it's interesting that the frequency of the appearance of waves is lower than in the data, I would have thought that the fitting of the neuromodulation period might have captured this slower time scale. Any comment on the discrepancy?

We thank the reviewer for highlighting this point. Indeed, the agreement between simulation and data concerning the frequency and the duration of down-states greatly improved after the addition of the acceptance criteria mentioned in previous answer R2.12. This is manifest in the novel version of Fig.5 bottom.

Figure 5: Dynamics comparison. Raster plot of the detected waves in experimental data (top), in simulated data without neuro-modulation (output of the inner loop, center), and in simulated data with optimal neuro-modulation (output of the outer loop, bottom). For visualization purposes, only 30s out of the 40s of the recording (mouse 1, trial 1) are reported. All the waves detected in the three scenarios are classified through a GMM approach; the four GMM modes are identified from fitting the collection of experimental + simulated waves of all trials. In this chunk specifically, the fourth mode (pink) is not found. Wave colours indicate the resulting modes of propagation.

R2.14. Fig. 6C and F pie charts, I like the idea of throwing the model and data in together to show that they are not trivially distinguished and hence comparable, but can the authors make it clearer how to see this 'the GMM is "fooled"' point? The three pie charts all appear to be fairly different.

The aim of the GMM validation analysis is to compare the simulation with the experimental dataset on observables not directly used to fit the model. Specifically, GMM validation is

applied to the spatio-temporal propagation pattern of each individual wave. To address this point, we added two information:

First, In Fig. 6 we added a fourth pie-chart showing that shuffled-sims fit worse.

Second, we added the “A posteriori validation of the simulated wave modes” section in **Supplementary Material**, including the following text (line 46) that aims at a quantification of the fit:

<<As a first validation analysis of the fidelity of the model, we forced the optimal simulation and the control (shuffled) simulation over the empirical (experiment-only fitted) modes. The fraction of detected modes is represented in Fig.S3D, K. We also measured how well these modes describe the simulated datasets. To do so, we split the experimental dataset in a training and a testing dataset, we fit the GMM over the training one only (75% of waves) and measured the log-likelihood associated with the classification of the testing experimental dataset (upper bound), the simulated waves datasets and the shuffled control dataset (lower 52 bound) as depicted in Table 1. The confidence in the representation of the simulated dataset is $88 \pm 2\%$ (mouse 1) and $85 \pm 1\%$ (mouse 2).>>

Figure 6: Validation of propagation modes. Analysis over the full set of 6 experimental trials and corresponding optimal simulations (for each of the two mice). Optimal simulations are selected trial-wise following the grid search approach, as described in section The Outer Loop. A and E. Direct comparison between experimental and simulated distributions of wave velocity (left panel), direction (central panel), and IWI (right panel). B and F. Fraction of each of the detected modes in the full collection (experiments + simulations, left), in the sub-collection of experiments only (center), and simulations only (right). C and G. Modes fraction of shuffled simulated waves classified by a GMM fit over shuffled simulation + experimental dataset (see Fig. S3 in Supplementary Material for a detailed description). Shuffled simulation: a posteriori validation of the simulated model. D and H. Centroids of the 4 gaussian distributions (modes) in the downsampled channel space describing the collection of experimental and simulated waves fitted with the GMM.

R2.15. Are the equations dimensionally correct? It appears that $f(v)$ is a voltage, then in Eq. 8, $\mu\tau$ has the same dimensions as $f(v)$ but is the product of a current and a time constant, so is a charge rather than a voltage.

We acknowledge that we improperly called I_{ext} a current. To fix this, we included the membrane capacitance C_m in eq. (4). The same we did for bW , which is now a current. Now, considering that the inferred parameters I_{ext}/C and bW/C_m are in [mV/ms] (as expressed in the updated version of Fig.3) and W adimensional (as a consequence of eq.(5)), μ is expressed in mV/ms. Therefore $\mu\tau$ is a voltage. We also specified in the text at line **430**.

R2.16. " $k_{\text{ie}} = k_{\text{ii}} = -25$ in our case" -- units?

k_{ie} and k_{ii} are expressed in mV. We specified it in the text, line **508**.

R2.17. The authors varied the number of clusters used in the GMM, it would be helpful to see how their performance measure depended on the number of clusters.

We added the following text at line **159**:

<< The number of modes fitted by the GMM is automatically identified using the Bayesian Information Criterion (BIC) relying on a likelihood maximization protocol [41]. For additional details see Supp. Mat. section Detecting the number of components in the GMM >>

Where we added Fig.S4 and the text at lines **72** in the Supplementary material:

<<This criterion provides an estimate for the goodness of the fit made by the GMM in terms of predicting the data we actually have. The lower is the BIC score, the better is the model to actually predict the available data (and, by extension, the true unknown distribution of all data). Specifically, the optimal number of Gaussian components to be considered is thus defined as the minimum one producing a plateau in the BIC gradient, as shown in Fig. S4.>>

Figure S4: Number of GMM components identified by the BIC criterion. Upper row: BIC score for the GMM method when fitting different numbers of components, for experimental data (blue) and experimental+simulated data (orange). Lower row: gradient of the BIC score from corresponding panels above. Left panels refer to mouse 1, right panels to mouse 2. Vertical dotted lines depict the optimal numbers of components identified for both datasets.

R2.18. "parameter T linearly ranging from 1.2s to 2s." -- Fig. 4 suggests a higher upper bound was used. And is period T hitting the upper limit in most of the epochs?

We thank the reviewer for the suggestion. We extended the grid search by investigating a wider parameter space (period ranging from 0 to 6.9, amplitude ranging from 0 to 6). Also, we added the following text at line **238**

<<On the whole set of waves, we computed the cumulative distributions (i.e. on the whole set of waves) of the three local observables characterizing the identified travelling waves (speed, direction, and IWI, already introduced in Section Characterization of the Cortical Slow Waves). The distance between cumulative distributions we chose is the Earth Mover's Distance (EMD, see Methods section for more details). EMD is separately evaluated over each of these observables (Fig. 4E) in a grid search. We then combined the three EMDs as in Eq. (20). The resulting "Combined Distance" is reported in Fig. 4D for a single trial, and is further constrained by an additional requirement that excludes the zone marked in gray: we reject those simulations with too long down-states and too short up-states, if compared to experimental distributions (see purple and yellow lines in Fig. 4, respectively).>>

Furthermore, we added details about the rejection criteria at the end of the <<*The Outer Loop: grid search and data-simulation comparison*>>, line **584**

<<Finally, we imposed two constraints on the simulation output, in order to reject biologically meaningless values for A and T : 1) most of the down-state duration distribution and 2) most of the wave duration distribution need to be both lower than their minimum values observed in experimental data (after having removed outliers; acceptance criteria: 98th and 95th percentile, respectively). This allowed us to exclude an entire region of non-interesting simulations and identify the optimal, biologically meaningful results.>>

Figure 4: Outer Loop. A. The second step of the inference method consists in comparing simulations and data, to optimize neuro-modulation amplitude and period hyper-parameters. Here, we have a bi-directional flow of information from data to model and the other way round. B. Direct comparison of the distributions of velocity (left panel), waves direction (central panel), and IWI (right panel) between data and simulation. The simulation is obtained selecting the optimal point (i.e. the minimum “Combined Distance” between experimental and simulated distributions) for the hyper-parameters grid search (amplitude and period of the neuro-modulation signal, see panel D). C. Same as panel B, but for a non-optimal point of the grid search (see panel D). D. Results of the hyper-parameters grid search on a single trial. “Combined Distance” between simulated and experimental distributions (see text for details) is plotted as a function of the two hyper-parameters (amplitude and period). The gray area depicts the simulations rejected because of too long down-states (yellow line) or too short up-states (purple line), if compared with experimental distributions. The green cross depicts the optimum value, while the orange cross is a non-optimum, in fact corresponding to the inner-loop case ($A=0$). E. EMD between experimental and simulated distributions of velocities (left), directions (center) and IWI (right), for the same single trial of panel D. F-G. Summary (across trials, and for the two different mice) of optimal hyper-parameter values obtained from the grid search, and their distribution (on the right). H. Summary for the span between minimal and maximal “Combined Distances” for all the trials of the two mice, for the grid search in the same ranges of amplitude and period values of panel D. Black points refer to “Combined Distance” from the inner-loop simulation. On the right, the histogram of the lower bound of the EMD span is shown.

R2.19. Labeling is too small in many of the figures.

We fixed them.

R2.20. There should be a space between a number and its units, and units shouldn't be italic.

We fixed it.

R2.21.

p17 last line, "2x2" -- \times not x.

Line **596** has been modified accordingly.

Reviewer #3 (Remarks to the Author):

This paper presents an intriguing data- and model-driven analysis of travelling waves of cortical activity in high-resolution in-vivo recordings of mouse brain obtained from wide-field calcium imaging. The work integrates cutting edge data analysis methods for wave detection with a novel "two loop" technique for model and data assimilation.

The paper is very well written and clearly presented. It will definitely be of substantial interest to experimentalists using high field imaging techniques as well as researchers using computational models. I have a few suggestions for the authors consider in their revisions:

Major:

R3.1. The model fitting looks interesting, yields plausible dynamics and is well motivated. But in terms of (perhaps future work), only one sort of dynamic is possible (periodically modulated non stationary GMM modes): The authors should outline next steps to allow different generative models to be posed and a means of identifying the best of the competing models of the data (also incorporating a complexity penalty as otherwise the most complicated model will likely win).

We believe that this question is related to question R3.5, for this reason we included an unique additional paragraph in the discussion at line **380**:

<<Generally speaking, there are strong evidences of oscillatory neuro-modulation currents, affecting the cortex and coming from deeper brain areas (such as the brain-stem), with a sub-Hertz frequency [63, 64].

However, we acknowledge that this shape for the neuro-modulation might have a limited descriptive power, and we plan to develop more general models. A possible choice can be that of a combination of many oscillatory modes or noisy processes such as Ornstein-Uhlenbeck and gaussian processes. However, the risk of this kind of models is overparametrization. This raises the necessity of a penalty to discourage a too large number of parameters and avoid over fitting (see e.g. Akaike Information Criterion [65]).

Finally, we stress once again that in the current work we only considered spontaneous dynamics. Then, we plan to go beyond spontaneous activity, aiming at modeling the dynamics of slow waves when the network receives input signals [66, 67], ranging from simple (local) electrical perturbations to proper sensory stimuli. The methodology to achieve this would be very similar to what presented in [68, 69, 70]. We present an example of the

response of the inferred network to a pulse stimulation in Supp. Mat. Section Simulation with pulse stimulation.>>

R3.2. The sanity check on page 8 + figure 6 is very nice but does seem a bit circular in the sense that the same data and model are used to generate the simulated data, then test/validate the mix of experimental and simulated data. Ideally, the model should be trained and tested on different data (for cross-validation) and also the performance of the method in detecting the mix should use a metric which differs from that used to optimise the model fit.

We re-organized Figures 6 and 8 (now 6 and S3) and expanded the section “A posteriori validation of the simulated wave modes” in supplementary material to address these questions.

Specifically, we fit the GMM over a training portion of the experimental dataset, and we measured the log-likelihood on the projection of the simulated dataset and a different testing experimental one (line **46** in Supplementary Material).

<<As a first validation analysis of the fidelity of the model, we forced the optimal simulation and the control (shuffled) simulation over the empirical (experiment-only fitted) modes. The fraction of detected modes is represented in Fig.S3D, K. We also measured how well these modes describe the simulated datasets. To do so, we split the experimental dataset in a training and a testing dataset, we fit the GMM over the training one only (75% of waves) and measured the log-likelihood associated with the classification of the testing experimental dataset (upper bound), the simulated waves datasets and the shuffled control dataset (lower bound) as depicted in Table 1. The confidence in the representation of the simulated dataset is $88 \pm 2\%$ (mouse 1) and $85 \pm 1\%$ (mouse 2).>>

Moreover, the simulation parameters are fit through the sequential application of the inner and outer loop (depicted in Figures 3 and 4). Specifically, the outer loop relies on the comparison between cumulative macroscopic measurements of local velocity, direction and inter wave interval. No information about the spatio-temporal pattern of each wave is taken into consideration at this stage. To further stress this concept we added the following text at **line 283**,

<<It is worth noting that the features fit by GMM are the spatio-temporal propagation pattern of each wave (i.e. a point in the spatially downsampled channel space). These features differ from those used by the outer loop (i.e. local velocity, direction and iwi cumulative distributions), thus providing a validation for the experiment-simulation comparison.>>

Figure 6: Validation of propagation modes. Analysis over the full set of 6 experimental trials and corresponding optimal simulations (for each of the two mice). Optimal simulations are selected trial-wise following the grid search approach, as described in section The Outer Loop. A and E. Direct comparison between experimental and simulated distributions of wave velocity (left panel), direction (central panel), and IWI (right panel). B and F. Fraction of each of the detected modes in the full collection (experiments + simulations, left), in the sub-collection of experiments only (center), and simulations only (right). C and G. Modes fraction of shuffled simulated waves classified by a GMM fit over shuffled simulation + experimental dataset (see Fig. S3 in Supplementary Material for a detailed description). Shuffled simulation: a posteriori validation of the simulated model. D and H. Centroids of the 4 gaussian distributions (modes) in the downsampled channel space describing the collection of experimental and simulated waves fitted with the GMM.

R3.3. A related test would be to mix up experimental and simulated data where the latter are generated in a manner that is a poor fit to the experiment – then use the approach to ensure the 2 (experimental and counter-factual simulations) are not mixed by the model inference technique.

We thank the reviewer for the suggestion. Accordingly, (see the novel Fig S2 panels B and F) we applied the GMM validation protocol to the output of a log-likelihood maximization optimization that fits two extra parameters: the period and amplitude of the neuromodulation. We show these results in the novel supplementary material section “*Inferring neuromodulation parameters through likelihood maximization*”. Here, we show that the macroscopic observables used to fit the outer loop are not correctly reproduced by this direct fit (making these simulations the kind of suggested “poor fit to the experiment”). Thus, we fit the GMM analysis of such simulations and experimental datasets: indeed, there is almost no overlap on the identified modes associated to the experimental datasets and the ones associated to the simulated one.

Figure S2: Inferring neuro-modulation parameters through likelihood maximization. A/E. Comparison of global metrics (waves velocities, directions and IWI) for data and simulation, when neuro-modulation parameters are inferred through likelihood maximization. B/F. GMM analysis to identify propagation modes. Comparison between experimental data and simulations. C/G. Histograms and scatter plot of amplitude and period neuro-modulation parameters inferred by likelihood maximization for each of the six trials. D/H. Rastergram of waves onset, showing the comparison of the dynamics between model and data for a single trial (red point in panels C/G.).

R3.4. Since the authors adopt an exponential synaptic kernel, I suggest they link their approach to neural field theory, which derives a PDF based on a very similar form (see e.g. Robinson, et al (1997). Propagation and stability of waves of electrical activity in the cerebral cortex. Physical Review E, 56(1), 826)

We added the following text at line **88**, which includes the suggested reference:

<<Elliptical kernels are often used in neural field theories to predict wave properties as a function of the connectivity parameters [Capone et al.],[Coombes et al.]; this is usually referred to as a direct approach, going from parameters to dynamics. In [Robinson et al.] the authors propose equations that include several properties of the cerebral tissue (2D structure, temporal delays, nonlinearities and others), retaining mathematical tractability. This allows to analytically predict global mode properties (such as stability and velocity of propagating wavefronts) for different geometries. Here, we propose a methodology to implement an inverse approach, from wave properties to model parameters.>>

[Capone et al.] Cristiano Capone and Maurizio Mattia. “Speed hysteresis and noise shaping of traveling fronts in neural fields: role of local circuitry and nonlocal connectivity”. In: Scientific reports 7.1 (2017), pp. 1–10.

[Coombes et al.] Stephen Coombes. “Waves, bumps, and patterns in neural field theories”. In: Biological cybernetics 93.2 (2005), pp. 91–108.

[Robinson et al.] Peter A Robinson, Christopher J Rennie, and James J Wright. “Propagation and stability of waves of electrical activity in the cerebral cortex”. In: Physical Review E 56.1 (1997), p. 826.

R3.5. I’m not convinced of the strictly periodic nature of neuromodulation (although it likely is adequate for the present purposes). Most likely neuromodulation is a heavy-tailed stochastic process, or even a weakly damped mean reverting process. Some discussion of other choices for the form of equation (18) should be considered

We believe that the answer provided to R3.1 covers also this point.

Minor:

R3.6.

The paper is in general very clearly written. However, the abstract is a bit idiosyncratic. I suggest rewording the abstract to something like:

“The development of novel techniques to record wide-field brain activity enables estimation of data-driven models from thousands of recording channels and hence across large regions of cortex. These in turn improve our understanding of the modulation of brain-states and the richness of traveling waves dynamics. Here, we infer data-driven models from high-resolution in-vivo recordings of mouse brain obtained from wide-field calcium imaging. We then assimilate experimental and simulated data through the characterization of the spatio-temporal features of cortical waves in experimental recording. Inference is built in two steps: an inner loop that optimizes a mean-field model by likelihood maximization, and an outer loop that optimizes a periodic neuro-modulation via direct comparison of observables that characterize cortical slow waves. The model reproduces most of the features of the non-stationary and non-linear dynamics present in the high-resolution in-vivo recording of the mouse brain. The proposed approach offers new methods of characterizing and understanding cortical waves for experimental and computational neuroscientists.

We thank the reviewer for the suggestion. We inserted the proposed abstract in the revised manuscript.

R3.7. “However, the activity produced by the model inferred from the inner loop results is found to be much more regular as compared to the experimental activity. For instance, the down state duration and the wavefront shape express almost no variability when compared to experimental recordings.” – as evidenced by what features in which figure/panel?

We added the following text at line 231.

<<This is illustrated by Fig. 5 and novel Sup. Mat Fig S6.>>

R3.8.

The text at the bottom of p7 should refer to Fig 4 (not Fig 3).

Yes, definitely. Thank you very much for having us noticing this mistake. We have corrected lines 232 and 234 accordingly.

R3.9. I think the Discussion should briefly canvass some of the prior empirical work on cortical travelling waves in experimental data: see some suggested references below (no need to cite all of them!!)

Muller, L., Reynaud, A., Chavane, F. & Destexhe, A. The stimulus-evoked population response in visual cortex of awake monkey is a propagating wave. *Nat. Commun.* 5, 3675 (2014).

Rubino, D., Robbins, K. A. & Hatsopoulos, N. G. Propagating waves mediate information transfer in the motor cortex. *Nat. Neurosci.* 9, 1549–1557 (2006).

Muller, L. & Destexhe, A. Propagating waves in thalamus, cortex and the thalamocortical system: experiments and models. *J. Physiol. Paris* 106, 222–238 (2012).

Townsend, R. G. et al. Emergence of complex wave patterns in primate cerebral cortex. *J. Neurosci.* 35, 4657–4662 (2015).

Hindriks, R., van Putten, M. J. A. M. & Deco, G. Intra-cortical propagation of EEG alpha oscillations. *Neuroimage* 103, 444–453 (2014).

Burkitt, G. R., Silberstein, R. B., Cadusch, P. J. & Wood, A. W. Steady-state visual evoked potentials and travelling waves. *Clin. Neurophysiol.* 111, 246–258 (2000).

Muller, L., Chavane, F., Reynolds, J. & Sejnowski, T. J. Cortical travelling waves: mechanisms and computational principles. *Nat. Rev. Neurosci.* 19, 255–268 (2018).

Sato, T. K., Nauhaus, I. & Carandini, M. Traveling waves in visual cortex. *Neuron* 75, 218–229 (2012).

Prechtl, J. C., Cohen, L. B., Pesaran, B., Mitra, P. P. & Kleinfeld, D. Visual stimuli induce waves of electrical activity in turtle cortex. *Proc. Natl Acad. Sci. USA* 94, 7621–7626 (1997).

Besserve, M., Lowe, S. C., Logothetis, N. K., Schölkopf, B. & Panzeri, S. Shifts of gamma phase across primary visual cortical sites reflect dynamic stimulus- modulated information transfer. *PLoS Biol.* 13, e1002257 (2015).

Townsend, R. G., Solomon, S. S., Martin, P. R., Solomon, S. G. & Gong, P. Visual motion discrimination by propagating patterns in primate cerebral cortex. *J. Neurosci.* 37, 10074–10084 (2017).

Benucci, A., Frazor, R. A. & Carandini, M. Standing waves and traveling waves distinguish two circuits in visual cortex. *Neuron* 55, 103–117 (2007).

We thank for the important suggestions, we cited three of them that seemed closer to our work.

Muller, L., Reynaud, A., Chavane, F. & Destexhe, A. The stimulus-evoked population response in visual cortex of awake monkey is a propagating wave. *Nat. Commun.* 5, 3675 (2014).

Burkitt, G. R., Silberstein, R. B., Cadusch, P. J. & Wood, A. W. Steady-state visual evoked potentials and travelling waves. *Clin. Neurophysiol.* 111, 246–258 (2000).

Muller, L., Chavane, F., Reynolds, J. & Sejnowski, T. J. Cortical travelling waves: mechanisms and computational principles. *Nat. Rev. Neurosci.* 19, 255–268 (2018).

REVIEWERS' COMMENTS:

Reviewer #1 (Remarks to the Author):

The authors have addressed all my concerns.
I consider that the paper is now suitable to be published on this journal.

Reviewer #2 (Remarks to the Author):

The authors have adequately addressed the reviewer comments.

Reviewer #3 (Remarks to the Author):

I am happy with the revisions and responses to my prior concerns.
I'm very impressed by the ambition and technical quality of this paper.